# Endophilin-A3 and Galectin-8 control the clathrin-independent endocytosis of CD166

Henri-François Renard[1,9,10 ✉], François Tyckaert[1,9], Cristina Lo Giudice [2], Thibault Hirsch[3], Cesar Augusto Valades-Cruz [4,5,6], Camille Lemaigre[1], Massiullah Shafaq-Zadah[4], Christian Wunder [4], Ruddy Wattiez[7], Ludger Johannes [4], Pierre van der Bruggen[3,8], David Alsteens [2,8] & Pierre Morsomme[1,10 ✉]

While several clathrin-independent endocytic processes have been described so far, their biological relevance often remains elusive, especially in pathophysiological contexts such as cancer. In this study, we find that the tumor marker CD166/ALCAM (Activated Leukocyte Cell Adhesion Molecule) is a clathrin-independent cargo. We show that endophilin-A3—but neither A1 nor A2 isoforms—functionally associates with CD166-containing early endocytic carriers and physically interacts with the cargo. Our data further demonstrates that the three endophilin-A isoforms control the uptake of distinct subsets of cargoes. In addition, we provide strong evidence that the construction of endocytic sites from which CD166 is taken up in an endophilin-A3-dependent manner is driven by extracellular galectin-8. Taken together, our data reveal the existence of a previously uncharacterized clathrin-independent endocytic modality, that modulates the abundance of CD166 at the cell surface, and regulates adhesive and migratory properties of cancer cells.

[1] UCLouvain, Louvain Institute of Biomolecular Science and Technology, Group of Molecular Physiology, Croix du Sud 4-5, B-1348 Louvain-la-Neuve, Belgium. [2] UCLouvain, Louvain Institute of Biomolecular Science and Technology, NanoBiophysics lab, Croix du Sud 4-5, B-1348 Louvain-la-Neuve, Belgium. [3] UCLouvain, de Duve Institute, Avenue Hippocrate 75, B-1200 Bruxelles, Belgium. [4] Institut Curie, PSL Research University, Cellular and Chemical Biology unit, U1143 INSERM, UMR3666 CNRS, 26 rue d'Ulm, F-75248 Paris Cedex 05, France. [5] Institut Curie, PSL Research University, Space-Time imaging of Endomembranes Dynamics Team, UMR144 CNRS, 26 rue d'Ulm, F-75248 Paris Cedex 05, France. [6] Inria Centre Rennes-Bretagne Atlantique, SERPICO Team, Campus Universitaire de Beaulieu, F-35042 Rennes, France. [7] UMons, Research Institute for Biosciences, Proteomics and Microbiology Lab, Place du Parc 20, B-7000 Mons, Belgium. [8] WELBIO, Wavre, Belgium. [9] These authors contributed equally: Henri-François Renard, François Tyckaert. [10] These authors jointly supervised this work: Henri-François Renard, Pierre Morsomme. ✉email: henrifran@gmail.com; pierre.morsomme@uclouvain.be

The biological relevance of clathrin-independent endocytic processes is not always clear, especially in the pathophysiological context of cancer. One reason is that they are still poorly defined mechanistically. This situation calls for a better characterization of the molecular players involved, be they elements of endocytic machineries or cargoes. Lately, key elements of these machineries have been identified in the BAR domain protein family, which are specialized in membrane curvature induction and recognition[1]. In particular, proteins from the Endophilin-A (endoA) subfamily were shown to control the clathrin-independent uptake of various cargoes, such as β-adrenergic receptors, IL-2 receptor[2,3], or bacterial toxins[4]. Owing to their high sequence similarity, the three endophilin isoforms A1–A3 are mostly considered as redundant.

CD166/ALCAM (Activated Leukocyte Cell Adhesion Molecule) is a transmembrane immunoglobulin-like protein present at the cell surface. It is involved in cell–cell contacts through interactions with its ligand CD6, present at the surface of T lymphocytes and other immune cells, as well as in homophilic interactions with other CD166 molecules present at the surface of adjacent cells[5]. It has functions in various processes, such as migration of monocytes across endothelia[6], trafficking of leukocytes into the central nervous system[7], or activation of T cells[8–10]. Importantly, CD166 is overexpressed in various types of tumors where its expression level/cell surface abundance correlates with poor prognosis[11–16]. More recently, CD166 has been identified as a cancer stem cell marker[12,17], and proposed as a target for immunotherapy[18–22]. Despite its apparent importance in cancer progression, the trafficking mechanisms regulating the homeostasis of CD166 at the cell surface remain unknown.

In this study, using proteomics and advanced imaging approaches, we demonstrate that CD166 is a clathrin-independent cargo. We show that the endoA3 isoform—but neither A1 nor A2—functionally associates with CD166-containing early endocytic carriers, and physically interacts with the cargo. Our data further demonstrate that the three endoA isoforms control the uptake of distinct subsets of cargoes. In addition, we provide strong evidence that the construction of endocytic sites from which CD166 is taken up in an endoA3-dependent manner is driven by extracellular galectin-8 according to the glycolipid–lectin (GL–Lect) mechanism. Taken together, our data reveal the existence of a previously uncharacterized clathrin-independent endocytic process controlled by galectin-8 and endoA3, essential for the downmodulation of the tumor marker CD166 at the cell surface. Importantly, we provide additional data suggesting that this endocytic modality regulates adhesive and migratory properties of cancer cells.

## Results

### Quantitative proteomics of cell surface upon CME inhibition.
In order to discover clathrin-independent cargo proteins acting in cancer, we performed a quantitative proteomic screening for proteins whose cell surface exposure was unchanged or even decreased upon inhibition of CME (Clathrin-Mediated Endocytosis; Supplementary Fig. 1a; Supplementary Data 1). As expected, canonical cargoes of CME, such as transferrin and LDL receptors, significantly accumulated at the cell surface (~3.0- to ~3.5-fold increase), while the level of some previously described cargoes using CIE (Clathrin-Independent Endocytosis, e.g., CD44) or both CIE and CME (e.g., EGF receptor) remained unchanged. These proteomic results were verified by Western blot analysis (Supplementary Fig. 1b, c). Surprisingly, the cell surface expression of several proteins was decreased, one of the strongest hits being CD166 (~2.8- to ~3.4-fold decrease; Supplementary Fig. 1a, d; Supplementary Data 1). CD166 (also termed ALCAM

for Activated Leukocyte Cell Adhesion Molecule) is a cell surface transmembrane immunoglobulin-like protein expressed by antigen-presenting cells. Although CD166 gene expression is decreased upon CME inhibition (Supplementary Fig. 1e), it does not explain the even stronger decrease in the protein at the plasma membrane that was observed in the proteomic analysis (Supplementary Fig. 1a, d; Supplementary Data 1). We therefore considered that an alternative clathrin-independent endocytic process responsible for CD166 uptake might be upregulated[23]. Hence, this hit appeared as a candidate of interest for further studies.

### CD166 is a clathrin-independent cargo.
To confirm that CD166 is a CIE cargo, we measured its uptake upon inhibition of CME, obtained by treating cells with siRNAs against μ2-adaptin, clathrin heavy chain, or dynamin-2. In all these conditions, the uptake of anti-CD166 antibody remained unaffected, while the endocytosis of transferrin was strongly inhibited (Fig. 1a–c). Similar results were obtained with acute inhibition of CME by the drug ikarugamycin (Supplementary Fig. 2a) or the knocksideways technique (Supplementary Fig. 2b). Hence, rather than the upregulation of CIE of CD166, we propose that CME inhibition switches the post-endocytic trafficking of CD166 from recycling to the lysosomal degradation route. Such a crosstalk between CIE and CME has previously been proposed for other clathrin-independent cargoes[24]. In addition, continuous co-incubation of cells with fluorescently labeled anti-CD166 antibody and transferrin clearly showed that both cargoes do not co-localize during the first 5 min of endocytosis (Fig. 1d). Similarly, anti-CD166 antibody did not present any obvious co-localization with μ2-adaptin or clathrin light chain over time, in contrast to transferrin (Supplementary Fig. 2c–h). To get deeper insight into CD166 endocytosis under unperturbed live-cell imaging conditions, and on the full 3D volume of cells, we performed lattice light-sheet microscopy-based experiments to follow the uptake of anti-CD166 antibody in genome-edited clathrin-mRFP cells. We quantified 639 endocytic events (Supplementary Movies 1 and 2). Highly sensitive detection, advanced event tracking, and statistical analysis of confined fluorescent spots over time (Supplementary Fig. 2i) indicated that 69.9% of CD166 uptake events did not co-localize with clathrin (Fig. 1e, f; Supplementary Movies 1 and 2). By contrast, 71.2% of transferrin uptake events were clathrin-positive (Fig. 1g, h). Altogether, these data clearly demonstrate that the tumor marker CD166 is largely and preferentially endocytosed in a clathrin- and dynamin-independent manner.

### Specific function of endoA3 in CIE of CD166.
In order to further characterize the molecular mechanisms of CD166 uptake, we focused on the endophilin-A (endoA) family of BAR domain proteins, which together with other members of the BAR domain protein superfamily, have already been implicated in CIE[2–4,25–27]. Interestingly, a strong inhibition of CD166 uptake was observed upon depletion of the A3 isoform of endophilin-A (endoA3) in HeLa cells, while no effect was observed upon depletion of the two other isoforms (Fig. 2a; Supplementary Fig. 3a). Using isoform-specific antibodies, we analyzed the expression level of each endoA in the various siRNA-treatment conditions (Supplementary Fig. 3b). Plotting the relative expression levels of each endoA isoform versus the amount of endocytosed CD166 in these various conditions indicated a strong and significant linear correlation with the expression of endoA3, but not with the two other isoforms (Fig. 2b: linear regressions: endoA1 ($R^2 = 0.2666$) and endoA2 ($R^2 = 0.06904$), slopes not significantly different from zero with $F$ tests; endoA3 ($R^2 = 0.9566$), slope significantly different from zero ($P < 0.0001$) with $F$ test). Similarly, a specific

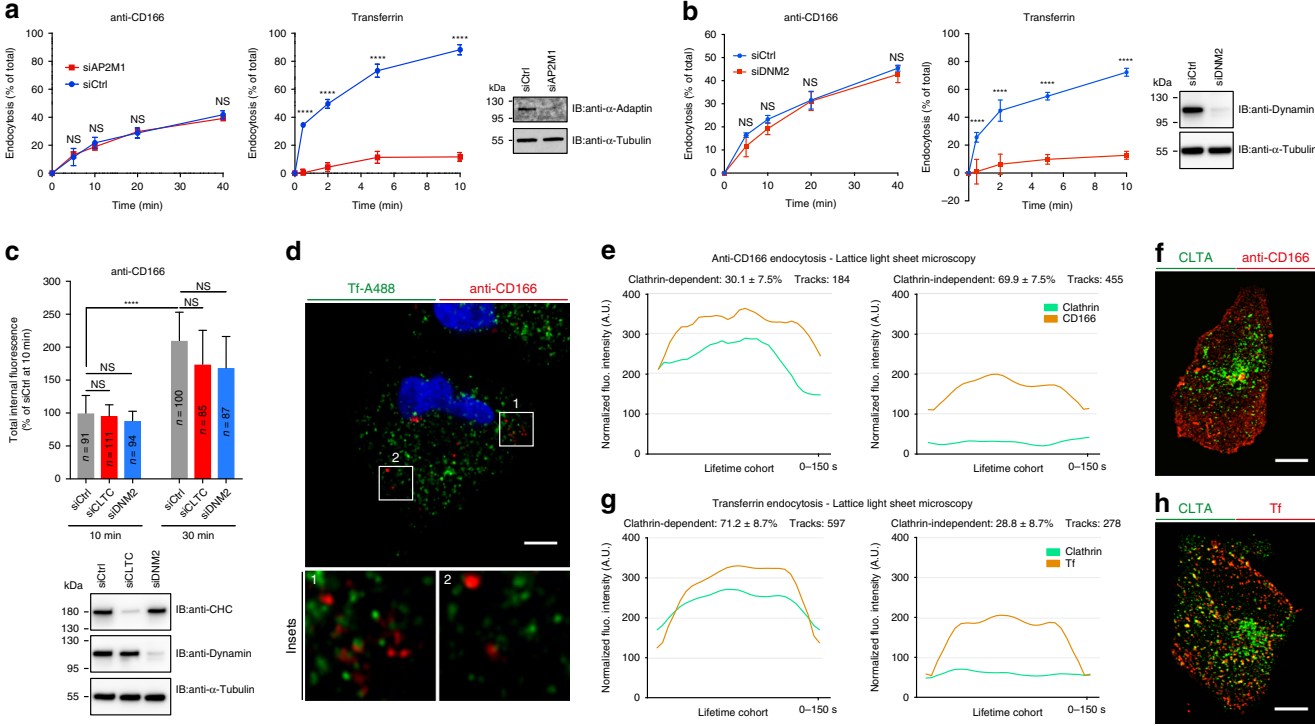

**Fig. 1 CD166 is endocytosed in a clathrin- and dynamin-independent manner.** HeLa cells treated with negative control (siCtrl, **a**–**c**), or μ2-adaptin (siAP2M1, **a**), dynamin-2 (siDNM2, **b**–**c**), or clathrin heavy chain (siCLTC, **c**) siRNAs. **a**, **b** Measure of anti-CD166 and transferrin (Tf) endocytosis by loss of surface assay in flow cytometry. **a** Number of independent experiments: anti-CD166, $n = 3$; Tf, $n = 4$. **b** Three independent experiments. NS, not significant. ****$P < 0.0001$ (RM two-way ANOVA with Bonferroni's multiple comparison test). **c** Continuous uptake of anti-CD166 antibody for 10 and 30 min. Internal fluorescence quantified from confocal images and plotted as the relative percentage of siCtrl condition at 10 min. $n$ cells, two independent experiments. NS not significant. ****$P < 0.0001$ (Kruskal–Wallis test with Dunn's multiple comparison post hoc test, two-sided). The Western blots against α-adaptin (**a**), dynamin (**b**, **c**), and clathrin heavy chain (**c**) document the efficiency of siRNAs. Loading control, α-tubulin. **d** Continuous co-uptake of anti-CD166 antibody (red) and fluorescent Tf (green) for 5 min in HeLa cells. No co-localization between the two markers (see insets). Representative of three independent experiments. **e**–**h** Endocytosis imaged by lattice light-sheet microscopy on live cells. Full 3D volume of 60 planes per U2OS cell acquired within 3 s. Average endocytic intensity traces per lifetime, and classification of endocytic events for anti-CD166 (**e**) and Tf (**g**) uptake. Green, endogenous genome-edited CLTA-mRFP; Orange, cargo. Lifetime cohorts of endocytic trajectories show a "dome"-shaped intensity profile of cargoes and co-tracking or not with clathrin. **f**, **h** 2D projections of acquired 3D stacks of cargo (red), clathrin (green), and overlay (yellow). Representative of ten experiments. All ten experiments were used for quantifications (**e**, **g**). Scale bars, 5 μm (**d**), 10 μm (**f**, **h**). Data are mean ± s.e.m. Source data are provided as a Source Data file and in Supplementary Fig. 11 (blots).

inhibition phenotype of CD166 uptake was observed upon endoA3 depletion in U2OS cell line (Supplementary Fig. 3c). In addition, overexpression of endoA3—but not endoA2—significantly increased the uptake of CD166 in HeLa cells (Supplementary Fig. 3d). Interestingly, exogenous expression of endoA3 in LB33-MEL cell line—that does not endogenously express endoA3 (see Supplementary Fig. 4e)—strongly restored CD166 uptake (Supplementary Fig. 3e). Of note, combined depletion of endoA3 and clathrin heavy chain in HeLa cells did not show further inhibition of CD166 uptake in our experimental conditions (Supplementary Fig. 3f). To generalize our findings, we tested several cell lines naturally expressing endoA3, and found that the cellular abundance of CD166 protein increased upon endoA3 depletion (Supplementary Fig. 4a–c: HeLa, HMC3, and U2OS, respectively). As expected, no change was observed in endoA3-negative cell lines (Supplementary Fig. 4d, e: SUM159 AP2-GFP, LB33-MEL, and MZ-2-MEL.43, respectively). This observation suggests a stabilization of CD166 at the cell surface in the absence of endoA3. This was confirmed by surface staining and flow cytometry analyses in HeLa and U2OS cell lines (Supplementary Fig. 4f, g), as well as by Western blotting on enriched cell surface protein fractions in HeLa cells (Supplementary Fig. 4h). Taken together, these data show that the endocytosis of

CD166 is specifically controlled by endoA3, but not endoA1 and endoA2, suggesting that the different isoforms have cargo-specific functions.

A closer observation of CD166 and endoA isoforms in super-resolution Airyscan confocal microscopy revealed the specific association of CD166-positive endocytic structures with endoA3, but not A1 or A2 isoforms in various cell types (Fig. 2c: HeLa; Supplementary Fig. 5a, b: BSC-1 and HMC3, respectively). This association was further confirmed by STED microscopy where early endocytic structures containing anti-CD166 antibody were clearly positive for GFP-tagged endoA3 (Supplementary Fig. 5c). Spinning-disk live-cell imaging provided a dynamic view of CD166 uptake process in endoA3-positive tubular invaginations emanating from the cell surface (Supplementary Fig. 5d, white arrows; Supplementary Movies 3 and 4). Some fission events could also be observed (Supplementary Movie 5). TIRF imaging of live cells showed that GFP-labeled endoA3 has a very short lifetime at the cell surface, as ~90% of spots lasted for less than 30 s (Supplementary Fig. 5e–g, green bars on the histogram). Interestingly, endoA3 became much less dynamic at sites of CD166 clustering, as ~80% of endoA3 spots had a lifetime exceeding 50 s and often the total duration of our 120-s movies (Supplementary Fig. 5e–g, red bars on the histogram; Supplementary Movie 6).

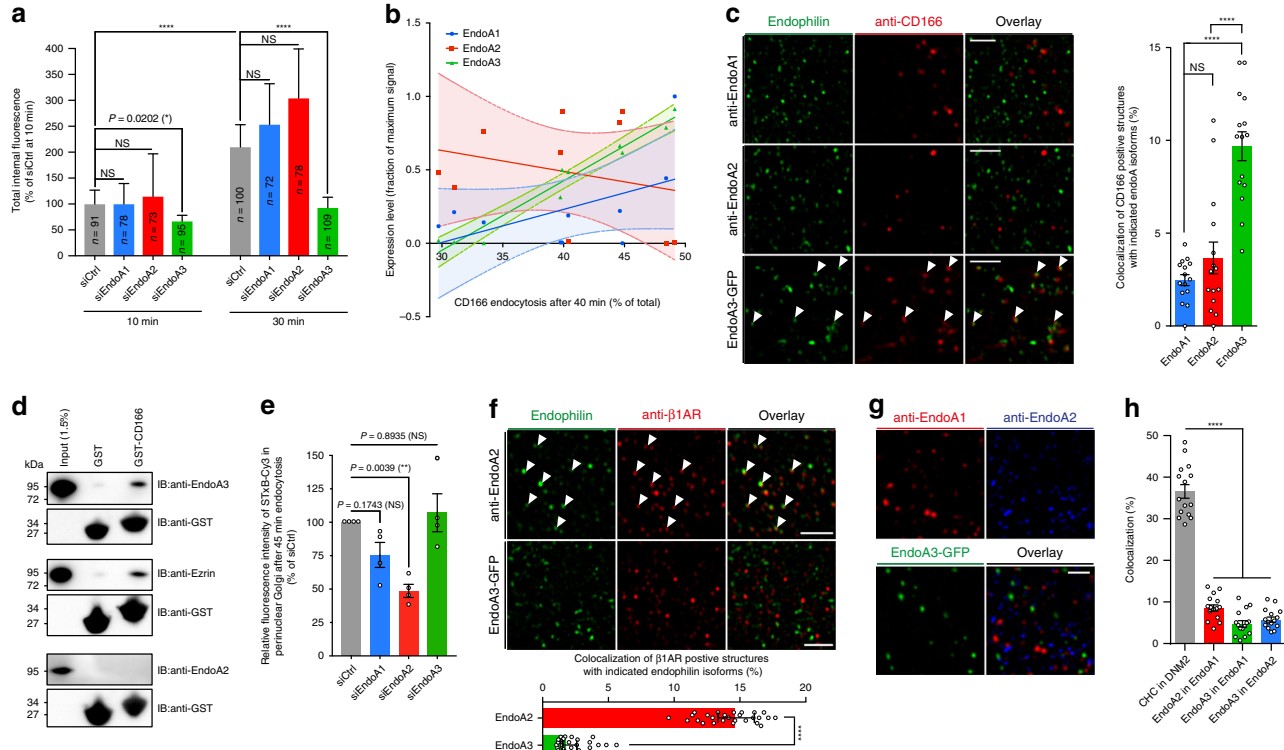

**Fig. 2 Endophilin-A3 controls the clathrin-independent uptake of CD166.** HeLa (**a–e**, **g**, **h**) or BSC-1 (**f**) cells treated with siRNAs or transfected with GFP constructs, as indicated. **a**, **b**, **e**, Treatment with negative control (siCtrl) or endoA1, A2, or A3 (siEndoA1, siEndoA2, or siEndoA3) siRNAs. **c**, **f–h**, Immunodetection of endogenous endoA1 and endoA2 with isoform-specific antibodies, or stable low-level expression of endoA3-GFP. **a** Continuous uptake of anti-CD166 antibody for 10 and 30 min. Internal fluorescence quantified from confocal images and plotted as the relative percentage of siCtrl condition at 10 min. n cells, two independent experiments. NS, not significant. *$P < 0.05$, ****$P < 0.0001$ (Kruskal–Wallis test with Dunn's multiple comparison post hoc test, two-sided). **b** Correlations between the percentage of anti-CD166 endocytosis (after 40 min, Supplementary Fig. 3a) and the relative expression level of individual endoA1 (blue), A2 (red), and A3 (green) isoforms (quantified from blots, Supplementary Fig. 3b). **c** Airyscan images of cells fixed after 10 min of incubation with anti-CD166 antibody (red). White arrows, co-localization. Graph, quantifications of co-localization ($n = 15$ cells, three independent experiments). NS, not significant. ****$P < 0.0001$ (ordinary one-way ANOVA with Bonferroni's multiple comparison test). **d** Pull-down experiments with cytosolic tail of CD166 on endoA3-GFP- or endoA2-GFP-expressing cell lysates. Negative control, GST alone. Positive control, ezrin. Representative of four independent experiments. **e** Incubation for 45 min at 37 °C with 50 nM STxB-Cy3. Fluorescence intensity in perinuclear Golgi quantified from tiled wide-field images, and normalized to siCtrl condition. Number of cells: siCtrl, $n = 5918$; siEndoA1, $n = 6147$; siEndoA2, $n = 4791$; siEndoA3, $n = 6290$. Four independent experiments. NS, not significant. **$P < 0.01$ (ordinary one-way ANOVA with Dunnett's multiple comparison test and a single pooled variance). Representative images, Supplementary Fig. 7a. **f** Uptake of β1-adrenergic receptor (β1-AR) after addition of 10 μM dobutamine for 4 min at 37 °C. β1-AR (red) and endoA2 or endoA3 (green). Graph, quantifications of co-localization ($n = 30$ cells, three independent experiments). ****$P < 0.001$ (two-tailed Mann–Whitney test). **g**, STED images of endoA isoforms (A1, red; A2, blue; A3, green). Equivalent confocal images, Supplementary Fig. 7c. **h**, Quantification of co-localization of endoA isoforms from confocal images (Supplementary Fig. 7c). Positive control, clathrin heavy chain, (CHC) and dynamin-2 (DNM2). Number of cells per condition: $n = 15$. Three independent experiments. ****$P < 0.0001$ (ordinary one-way ANOVA with Dunnett's multiple comparison test). Scale bars, 2 μm (**c**, **f**) and 1 μm (**g**). Data are mean ± s.e.m. (**c**, **e**, **h**), mean ± 95% CI (**b**), or median ± 95% CI (**a**, **f**). Source data are provided as a Source Data file and in Supplementary Fig. 11 (blots).

High endoA3 dynamics likely reflects the fact that the formation of endoA3-mediated endocytic carriers is aborted in the absence of cargo, while association with CD166 at uptake sites stabilizes the BAR domain protein for longer times. Similarly, very dynamic patterns were previously observed for endoA2 in the absence of cargoes, such as Shiga Toxin B-subunit (STxB)[4] and ligand-stimulated β-adrenergic receptors[3]. This specific association of endoA3 with CD166-containing endocytic structures suggested a possible physical interaction between both proteins. To address this question, we performed pull-down experiments from HeLa cell lysates using the cytosolic tail of CD166 as a bait. Interestingly, we could pull down endoA3, but not endoA2 (Fig. 2d). This interaction is apparently weak, which likely reflects a transient association of the two proteins. Of note, we could also pull down ezrin—used here as a positive control—for which the interaction with the cytosolic tail of CD166 has previously been documented[28].

Together with the functional analyses, these co-localization and physical interaction data further highlight the specific association of CD166 endocytosis with endoA3, but not endoA1 or endoA2.

**Distinct endocytic functions of endoA isoforms.** In order to further investigate these distinct endocytic functions of endoA isoforms, we performed experiments with previously described CIE cargoes whose uptake depends on endoA proteins. Previous studies have shown that endoA2 mediates the uptake of the bacterial STxB by controlling fission[4,29]. Others have found that CIE of endogenous cargoes, such as IL-2 or β-adrenergic receptors, is controlled by endoA proteins, which was termed Fast Endophilin-Mediated Endocytosis (FEME)[2,3]. Although these studies mainly focused on endoA2, they mostly consider the three endoA isoforms as redundant, which is understandable in view of

their high sequence similarity (Supplementary Fig. 6). First, we observed the intracellular transport of the B-subunit of STxB upon depletion of each individual endoA isoform (Fig. 2e; Supplementary Fig. 7a). As expected, the knockdown of endoA2 strongly inhibited retrograde STxB trafficking from the cell surface to perinuclear Golgi, while no effect was observed for the depletion of A3 isoform. Co-localization experiments indicated that the long plasma membrane invaginations induced by STxB were devoid of CD166 (Supplementary Fig. 7b). Likewise, co-uptake of the canonical FEME cargo β1-adrenergic receptor with anti-CD166 antibodies showed a poor overlap between the two markers (Supplementary Fig. 7c). By contrast with CD166 that clearly co-localized with endoA3 but not the A2 isoform, β1-adrenergic receptor revealed the exact mirror phenotype, i.e., co-localization with endoA2, but not endoA3 (Fig. 2f). Interestingly, the cellular localization of the three endoA isoforms, as observed by confocal, STED, or TIRF microscopy—using isoform-specific antibodies for endoA1 and endoA2, and a low expression of GFP-tagged endoA3—revealed a very poor overlap of the three patterns (Fig. 2g, h; Supplementary Fig. 7d, e). In addition, endoA3 showed very little co-localization with clathrin heavy chain or dynamin-2 (Supplementary Fig. 8a-d), and the lifetime at the plasma membrane for endoA3 spots (~10 s) was much shorter than that of the two CME markers (>20 s; Supplementary Fig. 8e, f; Supplementary Movies 7 and 8). Moreover, the uptake of transferrin was not affected by endoA3 depletion (Supplementary Fig. 8g, h). Together, these data demonstrate that endoA isoforms do not co-localize and control distinct endocytic processes for specific subsets of cargoes. More specifically, they underline the previously unrecognized role of endoA3 isoform in a distinct CIE mechanism, whose first identified cargo is CD166. Interestingly, specificity in this endocytic modality may come from direct interaction between the cytosolic tail of CD166 and endoA3 isoform.

**Role of galectin-8 in endoA3-dependent uptake of CD166.** Next, we investigated if extracellular factors could also control CD166 endocytosis in concert with endoA3. CD166 protein is heavily glycosylated, and previous reports have shown that it is a strong and specific binding partner of the mammalian lectin galectin-8 (Gal8)[30,31]. According to the glycolipid–lectin (GL–Lect) hypothesis, galectins co-cluster glycosylated cell surface cargo proteins and glycosphingolipids to drive the formation of tubular endocytic pits from which morphologically distinct endocytic structures termed CLICs (clathrin-independent carriers) are formed for the cellular uptake of these cargoes[32]. A previous study reported on the galectin-3 (Gal3)-driven CIE of CD44 and β1-integrin via this mechanism[33]. To address the role of galectins in the uptake of CD166, we first assessed the effect of N-Acetyl-D-lactosamine (LacNAc)—a strong competitor for the binding of galectins to carbohydrates on proteins or lipids—on the process. Interestingly, the uptake of anti-CD166 into cells is reduced by ~60% in the presence of 10 mM LacNAc in the culture medium (Supplementary Fig. 9a), suggesting that CD166 uptake is driven by galectins, Gal8 being the most likely candidate. In addition, we depleted glycosphingolipids by a 2-day incubation of cells with the glucosylceramide synthase inhibitor Genz-123346[34,35], and we observed an ~50% reduction of anti-CD166 uptake (Supplementary Fig. 9b, top-right panel). Of note, Gal3 and Tf uptake was used as controls of GL–Lect and non-GL–Lect cargoes, respectively (Supplementary Fig. 9b, bottom-left and bottom-right panels, respectively). Efficiency of Genz treatment was also verified by the loss of STxB binding to the cell surface (Supplementary Fig. 9b, top-left panel). Next, we performed anti-CD166 antibody uptake assays in the presence of purified

galectins. We observed a clear co-localization of Gal8 with anti-CD166 in endocytic structures (Fig. 3a). By contrast, co-localization of anti-CD166 with Gal1 (for which no specific interaction with CD166 could be detected[31]) was much lower and used as a negative control (histogram of Fig. 3a; Supplementary Fig. 9c). Of note, anti-CD166 did not show any significant co-localization with Gal3, a previously described marker of some CLIC populations[33,36] (Supplementary Fig. 9d). Strikingly, anti-CD166 antibody uptake was reduced by 50% in cells that were depleted of endogenous Gal8 by RNA interference, and cultured in serum-free medium (Fig. 3b; Supplementary Fig. 9e). The uptake of CD166 was rescued by the addition of exogenous purified Gal8 at doses as low as 0.2 nM (Fig. 3b). This observation is consistent with previously published concentration ranges of exogenous Gal3 required to rescue the uptake of CD44 on Gal3-depleted cells[33]. These nanomolar concentrations are also in line with the amount of Gal8 and other galectins found in the serum of healthy individuals and cancer patients[37,38]. Together, these data demonstrate that the endocytosis of CD166 is dependent on Gal8 and glycosphingolipids, a typical feature of endocytic processes that operate according to the GL–Lect hypothesis.

Next, we investigated the connections between Gal8 and endoA3. After 10 min of endocytosis in cells expressing GFP-tagged endoA3, we observed a significantly higher number of co-localization events with Gal8 than with Gal1 (Fig. 3c; Supplementary Fig. 9f). The same cells observed by TIRFM indicated a co-localization of Gal8 and endoA3 at the cell surface (Supplementary Fig. 9g). Of note, combined depletion of endoA3 and Gal8 did not further reduce CD166 uptake, suggesting that they are functionally involved in the same mechanism (Supplementary Fig. 9h). In order to scrutinize this association in a dynamic manner, we used a highly innovative device composed of a Fluid-FM—an instrument combining microfluidics and AFM (atomic force microscopy)—coupled to a fast-scanning confocal microscope (Fig. 3d). Gold nanoparticles functionalized with fluorescently labeled Gal8 or Gal1 were trapped at the apex of the Fluid-FM probe and subsequently approached to cells expressing GFP-tagged endoA3 in a force-controlled manner, thus achieving spatially and temporally resolved cell stimulation. Strikingly, we observed a significant increase in GFP signal at the plasma membrane around the nanoparticle within 60 s after contact only when it was coated with Gal8 (Fig. 3e, f; Supplementary Movie 9). Gal1-coated nanoparticles, uncoated nanoparticles (labeled with a control dye), or bare fluid-FM probes did not induce any significant recruitment of endoA3 to plasma membrane (Fig. 3f; Supplementary Fig. 9i–k; Supplementary Movies 10–12). Of note, we did not observe any recruitment of GFP-tagged endoA2 with Gal8-coated nanoparticles (Fig. 3f; Supplementary Fig. 9l; Supplementary Movie 13). These data indicate that extracellular Gal8 induces the fast and specific recruitment of endoA3 to uptake sites at the plasma membrane.

**Role of CD166/endoA3 in cancer cell adhesion and migration.** To place these findings in the broader context of cancer cell physiology, we decided to investigate how modulation of CD166 surface abundance by this new endocytic modality impacts two important features of cancer cells: migratory and adhesive properties. Previous studies have shown that a reduced expression of CD166 decreases cancer cell tumorigenicity and proliferation, while it favors migration[39]. In addition, several studies have shown that endoA3-expressing cancers have bad prognosis, as it is correlated with an increase in cancer cell proliferation and migration[40–42]. Recently, Poudel et al.[43] have shown that endoA3 promotes growth and migration of colon

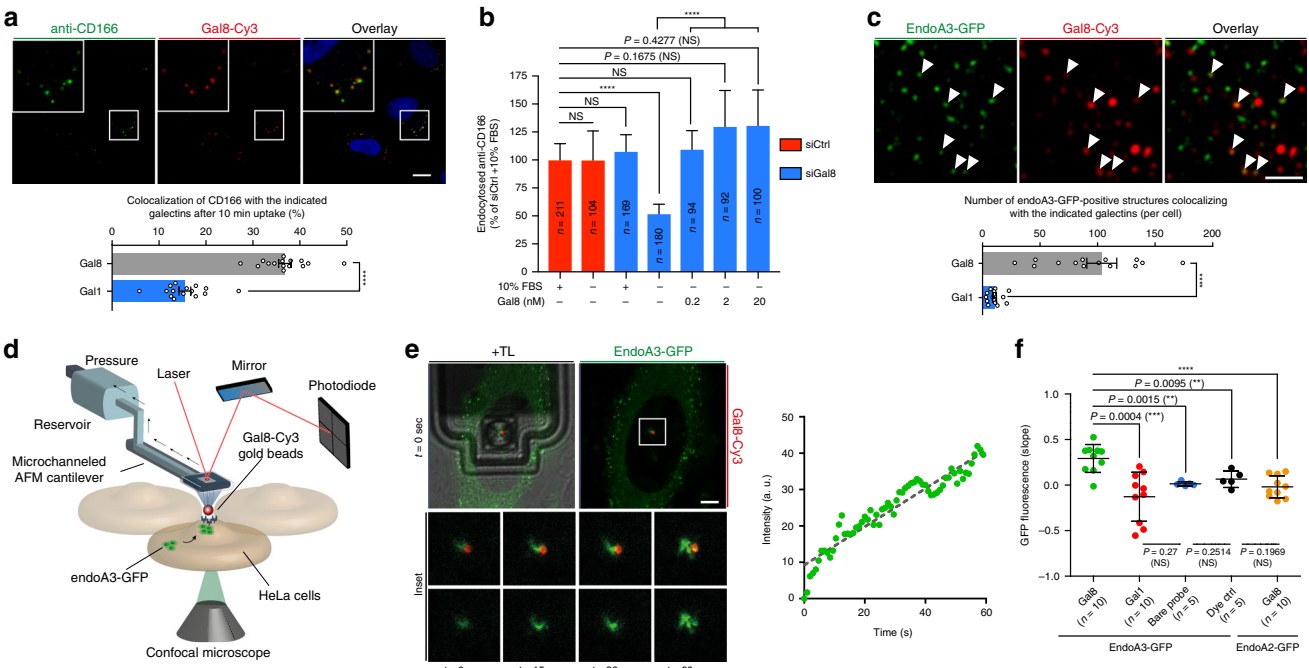

**Fig. 3 Galectin-8 drives the uptake of CD166 and induces the recruitment of Endophilin-A3 to endocytic sites at the plasma membrane.** HeLa cells. **a** Co-incubation with 5 µg ml[−1] anti-CD166 antibody and 50 nM fluorescent Galectin-8 (Gal8) for 10 min at 37 °C. Graph, quantifications of co-localization (n = 15 images with 5–15 cells per image). Three independent experiments. ****P < 0.0001 (two-tailed unpaired t test with equal variances). Galectin-1 (Gal1) condition, Supplementary Fig. 9c. **b** Cells treated with control (siCtrl, red) or Gal8 (siGal8, blue) siRNAs incubated with 5 µg ml[−1] anti-CD166 antibody for 10 min at 37 °C in various conditions (graph: ±10% FBS, ±purified Gal8). Intracellular accumulation of anti-CD166 quantified from confocal images. n cells, four independent experiments. NS not significant. ****P < 0.0001 (Kruskal–Wallis test with Dunn's multiple comparison post hoc test, two-sided). **c** Cells stably expressing endoA3-GFP incubated with 50 nM fluorescent Gal8 for 10 min at 37 °C. White arrows, co-localization. Graph, number of co-localization events per cell. Number of cells: Gal8, n = 35; Gal1, n = 31 cells. Three independent experiments. ****P < 0.0001 (two-tailed unpaired t test with Welch's correction). Gal1 condition, Supplementary Fig. 9f. **d–f** Gal8-induced recruitment of endoA3-GFP at plasma membrane. **d** Fluid-FM coupled to the confocal setup. Fluorescent galectin-coated gold nanoparticles trapped by microchanneled cantilever and approached to endoA3-GFP-expressing cells. Simultaneous monitoring of galectin (red) and endoA3 (green) fluorescence by fast-scanning confocal microscopy. **e** Representative images of Fluid-FM/confocal experiments (ten independent measurements). t = 0 s, initial cell exposure to Gal8. TL, transmitted light (cantilever). Insets: area surrounding the beads at the indicated time points. Graph: green fluorescence intensity around beads versus time. Images and plots for Gal1, dye control, bare probe, and endoA2-GFP controls, respectively, Supplementary Fig. 9i–l. **f** Quantification of fluorescence over time (slopes of linear regressions fitted on intensity vs. time curves). n-independent measures. NS not significant. **P < 0.01, ***P < 0.001 (two-tailed unpaired t tests). Scale bars, 10 µm (**a**), 1 µm (**c**), and 5 µm (**e**). Data are mean ± s.e.m. (**a**, **c**, **f**) or median ± 95% CI. **b** Source data are provided as a Source Data file.

cancer cells via two competing mechanisms: an endocytic mechanism that is required for proliferation, and the activation of Rac1 GTPase through interaction with TIAM1 GEF for cell migration. Impaired balance between those two mechanisms may favor hyperproliferative versus pro-metastatic phenotypes in vivo. Here, in addition, we propose that expression of endoA3 activates an endocytic mechanism that enables cancer cells to down-modulate cell surface proteins, such as CD166, which would allow them to tune their migratory and adhesive properties. This endocytic modality would further contribute to hyperproliferative and/or pro-metastatic phenotypes. To address this hypothesis, we analyzed the migration capacity of cancer cells in wound-healing assays. As expected, the depletion of endoA3 strongly reduced gap closure speed by a factor of 4, compared with the control condition in U2OS cell line (Fig. 4a, b; Supplementary Movie 14). Strikingly, in the absence of CD166, depletion of endoA3 did not have any additional effect on gap closure (Fig. 4a, b; Supplementary Movie 14). This result indicates that the migratory phenotype caused by the absence of endoA3 is tightly dependent on the presence of CD166. In order to extract more information from these wound-healing assays, we performed single-cell tracking analyses (Fig. 4c–e; Supplementary Fig. 10a). We clearly observed that endoA3 depletion strongly reduces the migration velocity of individual cells at the edge of the wound

(Fig. 4c, siEndoA3 condition; Fig. 4d, blue bar). Interestingly, the depletion of CD166 had little impact on the migration velocity (Fig. 4c, siCD166 condition; Fig. 4d, green bar). Furthermore, in the context where CD166 is absent, additional depletion of endoA3 did not further affect cell migration velocity (Fig. 4c, siCD166/siEndoA3 condition; Fig. 4d, red bar). Of note, in all conditions, the directionality of migration toward the wound (measured with xFMI, Fig. 4e) and the Euclidian distance (Supplementary Figure 10a) were significantly decreased by comparison with the control condition. Accumulated tracks (Fig. 4c) depict clearly the loss of directionality and distance traveled by cells when endoA3 is depleted (siEndoA3 condition), while it is almost exclusively a loss of directionality when CD166 is depleted, whether endoA3 is present or not (siCD166 and siCD166/siEndoA3 conditions). Western blot analyses on U2OS extracts showed that all siRNA depletions were remarkably efficient (Supplementary Fig. 10f). Altogether, these data further underline that the phenotype of endoA3 depletion requires the presence of CD166.

We performed additional wound-healing assays with other cancer cell lines that do not express endoA3 or CD166 (SUM159 wild type; Supplementary Fig. 10b, g; Supplementary Movie 15), or that express CD166 but not endoA3 (genome-edited clone of SUM159 expressing AP2-GFP; Supplementary Fig. 10c, g;

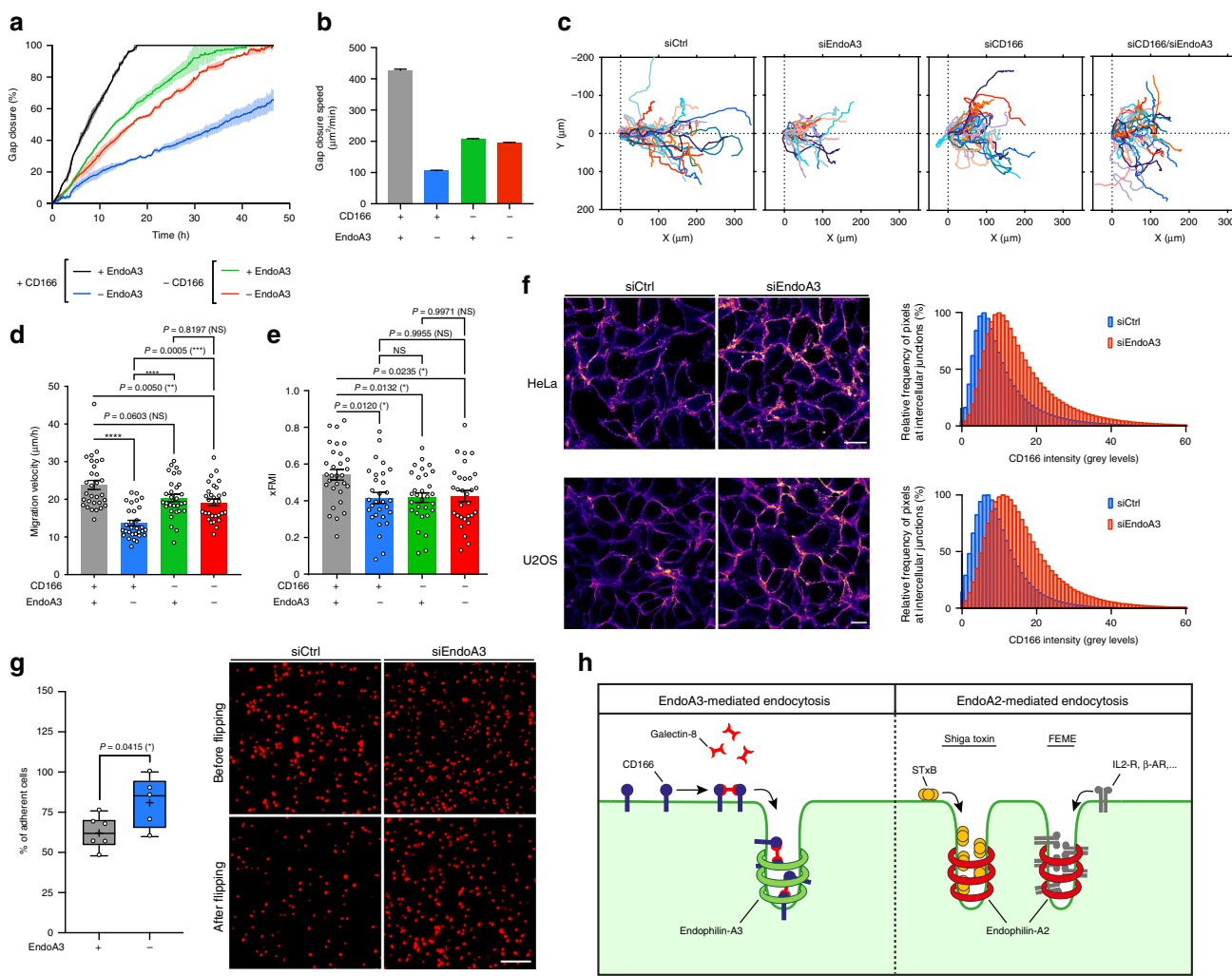

**Fig. 4 Modulation of CD166 cell surface abundance by endoA3-dependent endocytosis affects cancer cell adhesive and migratory properties.** U2OS (**a–f**) and HeLa (**f**, **g**) cells. **a–e** Wound-healing assay on cells treated with negative control (gray), endoA3 (blue), CD166 (green), or combined CD166 and endoA3 (red) siRNAs. See Supplementary Movie 14. **a** Gap closure over time (percentage of the initial gap area). Intervals, range between two technical replicates ($n = 2$). Representative of three independent experiments. **b** Average gap closure speed, extracted from linear regressions fitted to curves in (**a**). Representative of three independent experiments. **c–e** Single-cell tracking at the edge of the wound (800 min). **c** Accumulated tracks. **d** Average migration velocity. **e** Forward migration index along the x-axis (xFMI). $n = 30$ cells. Three independent experiments. NS not significant. *$P < 0.05$, **$P < 0.01$, ***$P < 0.001$, ****$P < 0.0001$ (ordinary one-way ANOVA, with Tukey's multiple comparison test). Euclidian distance, Supplementary Fig. 10a. **f** Accumulation of CD166 at intercellular junctions upon endoA3 depletion. Heatmap of anti-CD166 signal in confluent monolayers of cells treated with control (siCtrl) or endoA3 (siEndoA3) siRNAs. Histograms, quantification of junctional fluorescence intensity ($n = 5$ images). Representative of two independent experiments. **g**, Cell–cell adhesion flipping assay (Supplementary Fig. 10h; Methods). Adhesiveness of cells in suspension treated with control (siCtrl) or endoA3 (siEndoA3) siRNAs to a confluent monolayer of similarly treated cells, expressed as percentage of cells remaining attached to the monolayer after dish flipping (siCtrl, $n = 6$; siEndoA3, $n = 5$). Three independent experiments. *$P < 0.05$ (two-tailed unpaired t test). Additional data, Supplementary Fig. 10i. **h** Model. EndoA3/Gal8-mediated CIE is spatially and molecularly distinct from endoA2-mediated endocytosis. EndoA2 cargoes: bacterial toxins (e.g., STxB), G-coupled (β1-adrenergic, IL-2) receptors. EndoA3 cargo (this study): CD166. Scale bars, 20 μm (**f**), 200 μm (**g**). Data are mean ± range (**a**) or mean ± s.e.m. (**b**, **d**, **e**). Box plots (**g**) show median (−), mean (+), 25th and 75th percentiles (boxes), and data range from min to max (whiskers). Source data are provided as a Source Data file.

Supplementary Movie 16). As expected, transfection with siRNAs against endoA3, CD166, or a combination of both, had no significant impact on gap closure speed (Supplementary Fig. 10b, c). We also performed wound-healing assays with LB33-MEL cells, that do not express endoA3 endogenously, but express CD166. Interestingly, re-expression of exogenous endoA3 in LB33-MEL increased gap closure speed compared with wild-type cells (Supplementary Fig. 10d, e; Supplementary Movie 17). As a reminder, this re-expression of endoA3 also restored CD166 uptake (Supplementary Fig. 3e). Taken together, these data show that the strong effect on gap closure observed in U2OS cell line

upon endoA3 depletion is directly connected to the expression/function of the protein, and is not an off-target of siRNA treatment. More importantly, re-expression of endoA3 increases migration capacities of cancer cells that do not express the protein naturally, and this can be correlated with a restoration of CD166 uptake.

Upon depletion of endoA3 (where cells dramatically lose migration velocity and directionality, which in turn strongly reduces gap closure speed in wound-healing assays, see Fig. 4a–e), we observed that CD166 accumulates at the cell surface (Supplementary Fig. 4f–h). Interestingly, we could show in both

HeLa and U2OS cell lines that this accumulation happens at intercellular contacts (Fig. 4f). We next aimed to examine if modulation of CD166 surface level by endoA3-dependent CIE could affect intercellular adhesion. In order to address this question, we tested cell–cell adhesion using a previously described flipping assay[44,45], in which cells in suspension adhere to a monolayer of adherent cells (Supplementary Fig. 10h; see Methods section for more information). First, we observed the adhesiveness of cells treated with negative control siRNA, or siRNAs against CD166, endoA3, or a combination of both, to a wild-type (untreated) confluent cell layer (Supplementary Fig. 10i). This experiment clearly showed that a lack of CD166 significantly decreases intercellular adhesions. In a second experiment, we compared the adhesiveness of cells treated with siRNAs to a confluent layer of cells treated with the same siRNAs (Fig. 4g). Interestingly, we observed a significant increase in intercellular adhesiveness upon endoA3 depletion, as compared with control condition. These variations of intercellular adhesion correlate with the variations of CD166 abundance at the cell surface.

In other words, upon endoA3 depletion, the higher amount of CD166 at the cell surface increases intercellular adhesiveness, which results in slower movements and loss of directionality of individual cells. This may explain why gap closure in wound-healing assays is strongly impaired in the absence of endoA3 compared with control condition in U2OS cells. On the other hand, in the absence of CD166, the reduced intercellular interactions only result in a loss of directionality, with a very minor effect on migration velocity of individual cells. This may explain why gap closure is only slightly affected in these conditions. However, the very strong effect of endoA3 depletion on cell migration requires the presence of CD166. We conclude that endoA3-dependent CIE downmodulates CD166 from the cell surface, which reduces intercellular adhesion. Conversely, when endoA3 is depleted, CD166 accumulates at cell–cell junctions, resulting in stronger intercellular adhesion and reduced migration capacity of cells. Altogether, these observations indicate that fine-tuning of CD166 at the cell surface by endoA3-dependent CIE can be used by cancer cells to modulate intercellular adhesiveness, and ensure optimal collective migration with the highest speed and directionality.

## Discussion

Our study challenges the current prevailing view that the three endophilin-A isoforms are functionally redundant in CIE. Our observations reveal the existence of a previously uncharacterized clathrin-independent endoA3-dependent endocytic process, which should be distinguished from endoA2 and the previously described FEME pathway[2–4] (Fig. 4h). The data presented here provide strong evidence that endoA isoforms have distinct localizations at the plasma membrane and control the uptake of different subsets of cargoes from the cell surface. We have demonstrated that the tumor marker ALCAM/CD166 is a CIE cargo that is internalized in an endoA3-dependent manner. Pull-down data indicated that the specificity of endoA3 in the uptake of CD166 arises from physical interaction, although we cannot exclude the role of microdomains/lipids in the recruitment mechanism of the BAR domain protein. The exact mechanism through which endoA3 controls CD166 uptake remains unclear. Further insights into these mechanistic aspects will require more investigations, such as by mapping of interaction surfaces and generation of mutants disrupting this association. In addition, we identified Gal8 as a driver for the recruitment of endoA3 to uptake sites at the plasma membrane, and the endocytosis of CD166. The mammalian lectin Gal3 was previously proposed to promote the formation of endocytic invaginations by clustering glycosylated cargoes (CD44 and integrins) and glycosphingolipids at the cell surface[33], termed the GL–Lect mechanism[32]. This mechanism also operates in the case of Gal8-driven uptake of CD166, as we provided data showing that glycosphingolipids play a crucial role here. Our study is the first to identify a BAR domain protein module—namely endoA3—that directly acts on a galectin-induced endocytic process. It should be noted that the process we describe here shares striking similarities with the GL–Lect-dependent uptake of the bacterial lectin Shiga toxin[46], which induces the formation of plasma membrane invaginations that are recognized and processed by endoA2[4].

Additional investigation will be necessary to see if endoA3 also enables friction-driven scission, as described for endoA2[4,29]. Furthermore, it should be determined which other cargoes require endoA3 for their uptake, and whether other galectins also drive these endoA3-dependent uptake events. Finally, our study integrates endoA3/Gal8/CD166 endocytosis in a broader biological context, as the data suggest that it controls the homeostasis of CD166 at the cell surface in cancer cells, which can directly modulate their ability to adhere to neighboring cells and to migrate. Overall, our findings seem to indicate that cancer cells are able to tune their own migratory properties through the activation of specific CIE machineries. Nevertheless, the identification of additional endoA3/Gal8 cargoes will help to better understand the role of this endocytic modality in the behavior of cancer cells. In particular, it will be important to elucidate how Gal8 and other galectins interfere with cell adhesion and migration when CD166 is expressed. Based on our findings, future research in physiologically relevant in vivo models should help to decipher the role of this endocytic modality in cancer development.

## Methods

**Antibodies and other reagents**. The following antibodies were purchased from the indicated suppliers: mouse monoclonal anti-CD166 clone 3A6 (Bio-Rad, MCA1926, 1:200 for immunofluorescence and flow cytometry on human cell lines); goat polyclonal anti-CD166 (R&D Systems, AF1172, 1:20 for immunofluorescence on mouse cell lines); mouse monoclonal anti-CD166 clone B-6 (Santa Cruz Biotechnology, sc-74558, 1:500 for Western blotting); mouse monoclonal anti-TfR (BD Biosciences, 555534, 1:1,000 for Western blotting); rabbit polyclonal anti-α-adaptin (Santa Cruz Biotechnology, sc-10761, 1:1000 for Western blotting); rabbit monoclonal anti-EGFR (Cell Signaling Technology, 4267, 1:4000 for Western blotting); rabbit monoclonal anti-endoA1 (Cell Signaling Technology, 65469, 1:100 for immunofluorescence and 1:1000 for Western blotting); mouse monoclonal anti-endoA2 (Santa Cruz Biotechnology, sc-365704, 1:50 for immunofluorescence and 1:500 for Western blotting); mouse monoclonal anti-endoA2 coupled to Alexa Fluor 488 (Santa Cruz Biotechnology, sc-365704 AF488, 1:50 for immunofluorescence); rabbit polyclonal anti-endoA3 (Sigma Life Sciences, HPA039381, 1:1,000 for Western blotting); rabbit monoclonal anti-giantin (Institut Curie, recombinant proteins platform, A-R-R#05, 1:50 for immunofluorescence); mouse monoclonal anti-dynamin (BD Biosciences, 610245, 1:4,000 for Western blotting); mouse monoclonal anti-clathrin heavy chain for Western blotting (BD Biosciences, 610500, 1:5000); mouse monoclonal anti-α-tubulin (Sigma, T5168, 1:5000 for Western blotting); rabbit polyclonal anti-β1-adrenergic receptor (Abcam, ab3442, 1:1000 for immunofluorescence); rabbit polyclonal anti-Galectin-8 (Biorbyt, orb216142, 1:1000 for Western blotting); rabbit monoclonal anti-ezrin (R&D Systems, MAB72391, 1:1000 for Western blotting); mouse monoclonal anti-GST (Invitrogen, MA4-004, 1:1000 for Western blotting); unconjugated secondary antibodies or conjugated to Alexa Fluor 488, 546, or 647 (Thermo Fisher Scientific); anti-mouse and anti-rabbit secondary antibodies conjugated to horseradish peroxidase (Sigma and Dako, respectively). The mouse monoclonal anti-clathrin heavy chain antibody X22 (used at 1:50 for immunofluorescence) was a gift from E. Smythe. Of note, the specificity of anti-endoA1 and anti-endoA2 used above for immunofluorescence experiments was verified using cells depleted for each individual endoA isoform. 2-deoxy-D-glucose (D8375), sodium azide (S2002), ikarugamycin (SML0188), human transferrin–biotin conjugate (T3915), and dobutamine hydrochloride (D0676) were purchased from Sigma. Pierce Cell Surface Protein Isolation Kit (89881), human transferrin-Alexa Fluor 488 conjugate (T13342), rapamycin (PHZ1235), and HCS CellMask Deep Red (H32721) were purchased from Thermo Fisher Scientific. N-acetyl-D-lactosamine (OA08244) was purchased from Carbosynth. Genz-123346 compound (HY-12744A) was purchased from MedChemExpress.

**Cell culture**. HeLa (human cervix adenocarcinoma, ATCC® CCL-2), MEF[33,36] (mouse embryonic fibroblasts), BSC-1 (green monkey normal kidney, ATCC® CCL-26), U2OS (human osteosarcoma, ATCC® HTB-96), and genome-edited U2OS CLTA-mRFP, HMC3 (immortalized human microglia, ATCC® CRL-3304), SUM159 wild-type, and genome-edited SUM159 AP2-eGFP (human breast carcinoma) were grown at 37 °C under 5% CO$_2$ in DMEM high glucose glutamax (Gibco, 61965-059) supplemented with 10% FBS, 100 U ml$^{-1}$ penicillin, 100 µg ml$^{-1}$ streptomycin, and 1 mM pyruvate. Genome-edited SUM159 AP2-eGFP and U2OS CLTA-mRFP cell lines were gifts from T. Kirchhausen (Harvard Medical School) and D. Drubin (University of California, Berkeley), respectively. HeLaM cells stably expressing Mito-YFP-FRB and α-adaptin-FKBP were provided by M. S. Robinson (Cambridge Institute for Medical Research), and were grown at 37 °C under 5% CO$_2$ in DMEM high glucose glutamax (Gibco, 61965-059) supplemented with 10% FBS, 100 U ml$^{-1}$ penicillin, 100 µg ml$^{-1}$ streptomycin, 1 mM pyruvate, and 137.5 µg ml$^{-1}$ hygromycin B. Human cell line LB33-MEL was derived in 1988 from melanoma patient LB33[47]. This cell line was cultured in Iscove Medium (Gibco, 12440-053) supplemented with 10% FBS, 0.55 mM L-Arginine, 0.24 mM L-Asparagine, 1.5 mM Glutamine (AAG), 0.75 mM β-Mercaptoethanol, 100 U ml$^{-1}$ penicillin, and 100 µg ml$^{-1}$ streptomycin, and was grown at 37 °C under 8% CO$_2$. Human cell line MZ-2-MEL.43 was derived from melanoma patient MZ-2[48]. This cell line was cultured in DMEM supplemented with 10% FBS, 0.55 mM L-Arginine, 0.24 mM L-Asparagine, 1.5 mM Glutamine (AAG), 100 U ml$^{-1}$ penicillin, and 100 µg ml$^{-1}$ streptomycin, and was grown at 37 °C under 8% CO$_2$. HeLa and LB33-MEL cells stably expressing endoA3-GFP were generated for this study (see below), and grown in the same medium as the mother cell line, supplemented with 0.125 mg ml$^{-1}$ and 0.7 mg ml$^{-1}$ G418, respectively.

**DNA constructs and transfection**. Expression plasmids for C-terminally GFP-tagged rat endoA2 (P. De Camilli, Yale School of Medicine) and human endoA3 (H.T. McMahon, MRC, Cambridge), mCherry-tagged µ2 subunit of AP2 (C. Merrifield, CNRS—i2BC, Gif-sur-Yvette), and mRFP-tagged dynamin-2 (T. Kirchhausen, Harvard Medical School), were kindly provided by the indicated colleagues.

In order to visualize endoA3 in microscopy, a C-terminally GFP-tagged human endoA3 construct was engineered with the aim to generate a stable HeLa cell line. For this, human endoA3 sequence was amplified by PCR from the previous expression plasmid (forward primer with AflII restriction site: 5′-GCGCTTAAG ATGTCGGTGGCCGG-3′; reverse primer with EcoRI restriction site: 5′-CCGG AATTC GAAGCTTCTGAGGTAAAGGCACGA-3′). To obtain the final construct, rat endoA2 and human endoA3 sequences were swapped in the bicistronic vector pIRESneo2 encoding endoA2-GFP-FKBP generated in our previous study[4] between AflII and EcoRI restriction sites. Correct clones were validated by sequencing.

For pull-down assays, a GST-tagged CD166 tail construct was generated. CD166 cytoplasmic tail sequence was amplified by PCR from a C-terminally GFP-tagged human CD166 expression plasmid (J. Weiner, University of Iowa) (forward primer with EcoRI restriction site: 5′-GGAATTCTACATGAAGAAGTCAAAG-3′; reverse primer with XhoI restriction site: 5′-CCGCTCGAGTTAGGCTTCAGTT TTGTG-3′). To obtain the final construct, amplified CD166 tail sequence was inserted in the pGEX-4T1 expression plasmid (GE Healthcare) encoding for GST between EcoRI and XhoI restriction sites. Correct clones were validated by sequencing.

For immunofluorescence and live-cell imaging experiments, plasmids were transfected with FuGene HD (Promega) in HeLa and HCM3 cells, or with Lipofectamine 2000 (Thermo Fisher Scientific) in BSC-1 cells, according to the manufacturers' instructions. Cells were used for experiments 16–24 h after transfection.

For the production of HeLa and LB33-MEL cell lines stably expressing endoA3-GFP, pIRESneo2/endoA3-GFP–FKBP expression plasmid was transfected by nucleofection (Amaxa Nucleofector 2b, Lonza) and electroporation (BTX Gemini X2 System), respectively. A selection pressure was imposed with 0.8 and 0.7 mg ml$^{-1}$ G418 during 7–15 days for HeLa and LB33-MEL cell lines, respectively. At the end of the selection process, stable HeLa endoA3-GFP clones were isolated by single-cell sorting based on GFP fluorescence using FACSAria III flow cytometer (BD Biosciences). For stable LB33-MEL endoA3-GFP cell line, the total cell population resistant to G418 was recovered (no clone selection), and flow cytometry measurements indicated that 100% of cells were GFP-positive. Stable HeLa and LB33-MEL endoA3-GFP cell lines were then maintained in the continuous presence of 0.125 and 0.7 mg ml$^{-1}$ G418, respectively.

**RNA interference**. siRNAs used in this study were purchased from Qiagen and transfected with HiPerFect (Qiagen) according to the manufacturer's instructions, except for BSC-1 cells that were transfected twice (on day 0 and day 1) to reach efficient knockdown. Experiments were always performed 72 h after siRNA transfection, where protein depletion efficiency was maximal as shown by immunoblotting analysis with specific antibodies (routinely 80–100%). For most experiments, cells were replated 24 h before use, according to the needs of the experiment. AllStars Negative Control siRNA served as a reference point. The depletion of each endoA isoform was achieved with two different sequences, used

as a pool or individually, at a total final concentration of 20 or 40 nM: endoA1 #1 (SI04149411: 5′-AAAGACTCTTTGGACATAGAA-3′), endoA1 #2 (SI04260949: 5′-AAATCTGGTATCCAAGCTTA A-3′), endoA2 #1 (SI03057250: 5′-CACCAGC AAGGCGGTGACAGA-3′), endoA2 #2 (SI03073931: 5′-CATGCTCAACACGGT GTCCAA-3′), endoA3 #1 (SI04170376: 5′-CCAGACGAGAATACAAGC CAA-3′), and endoA3 #2 (SI04176529: 5′-CCAGACGAAGAAGTCAGACAA-3′). For µ2-adaptin depletion, a single siRNA sequence at a final concentration of 20 or 40 nM was used (SI02777355: 5′-TGCCATCGTGTGGAAGATCAA-3′). For galectin-8, a smartpool of four siRNA sequences at 40 nM was used: Gal8 #1 (SI04336612: 5′-AACCCACGCCTGAATATTAAA-3′), Gal8 #2 (SI04340028: 5′-TCGGACTTA CAAAGTACCCAA-3′), Gal8 #3 (SI03237339: 5′-TGGACGAACGTGTCGTCGTT AA-3′), and Gal8 #4 (SI04356506: 5′-AAGGTTGCAGTAAATGGCGTA-3′), except for Supplementary Fig. 9h where only sequences #1 and #2 were used; for clathrin heavy chain, a smartpool of four siRNA sequences at 40 nM (SI00299873: 5′-AAGGAGAGTCTCAGCCAGTGA-3′; SI00299880: 5′-TAATCC AATTCGAA GACCAAT-3′; SI04152372: 5′-AAGGGCTAACGTCCCAAATAA-3′; SI04190417: 5′-CCCTGAGTGGTTAGTCAACTA-3′); for dynamin-2, a smartpool of four siRNA sequences at 40 nM (SI02654687: 5′-CTGCAGCTCATCTTCTCAAAA-3′; SI03233671: 5′-TCGCAACCTGGTGGACTC ATA-3′; SI04224591: 5′-CTCATAC GTGGCCATCATCAA-3′; SI04277546: 5′-CCCGCTGGTCAAC AAACTGCA-3′); for ALCAM/CD166, three siRNA sequences were used at a final total concentration of 20 or 40 nM: CD166 #1 (SI02780162: 5′-CACCTGCTCGGTGACATAT TA-3′), CD166 #2 (SI02780169: 5′-AAGCATGAACGTGGATTGTAT-3′), CD166 #3 (SI03030286: 5′-AACGTGTTTGAGGCACCTAC A-3′). For the analysis of CD166 in cell lysates (Supplementary Fig. 4), a smartpool with the three sequences was used. For wound-healing assays (Fig. 4a–e and Supplementary Fig. 10a–c, f, g), a smartpool with sequences #1 and #2 only was used.

**Recombinant protein production**. Recombinant wild-type STxB was purified from bacterial periplasmic extracts as previously described[49]. For the purification of His-tagged recombinant human Gal1, Gal3, and Gal8, see Supplementary Methods.

**Recombinant protein and antibody labeling**. For further details, see Supplementary Methods.

**Quantitative mass spectrometry analysis of cell surface**. Cell surface proteins were isolated using the Pierce Cell Surface Protein Isolation Kit (Thermo Fisher Scientific) and digested by trypsin. Peptides were labeled with iTraq reporters and subjected to quantitative proteomic analysis. For further details, see Supplementary Methods.

**Quantification of gene expression by qPCR**. For further details, see Supplementary Methods.

**Light microscopy**. For immunofluorescence studies, cells were fixed with 4% paraformaldehyde at 37 °C for 10–20 min (Figs. 2c, e–h; 3c, 4f; Supplementary Figs. 2c–h, 5a–c, 7a, b, d, e, 8a–d, 9f) or on ice for 10 min followed by an additional 10 min at room temperature (Figs. 1c, d, 2a, 3a, b; Supplementary Figs. 3c, e, f, 7c, 8h, 9a–d, h), depending on the requirements of each experiment. After quenching with 50 mM NH$_4$Cl for at least 15 min and permeabilization with saponin (0.02% saponin, 0.2% bovine serum albumin (BSA) in phosphate-buffered saline (PBS)) for 30 min, both at room temperature, cells were incubated with primary and secondary antibodies for 30–60 min, and mounted with Fluoromount G (Invitrogen).

Fixed samples were imaged with a 34-channel Zeiss LSM 710 confocal microscope equipped with Airyscan system and with a Plan Apo 63× numerical aperture (NA) 1.4 oil immersion objective. Wide-field images were acquired on a Zeiss Axio Observer 7 epifluorescence microscope equipped with a C-Apo 40× NA 1.2 water immersion objective and an ORCA-Flash 4.0 LT sCMOS camera (Hamamatsu). STED (stimulated emission depletion) super-resolution images were acquired on an easy3D STED microscope (Abberior Instruments GmbH) equipped with 100× NA 1.4 oil immersion objective. Plasma membrane images of fixed samples were also acquired by TIRF microscopy, using a Nikon TIRF Ti-E inverted microscope equipped with a CFI Apo TIRF 100× NA 1.49 oil immersion objective and an EMCCD Evolve camera (Photometrics). Of note, those specific samples were conserved and imaged in PBS after immunofluorescence, rather than Fluoromount G that is not suitable for TIRFM.

For live-cell imaging, cells were grown to subconfluence on chambered glass coverslips (µ-slides with glass bottom, Ibidi). Observations were made at 37 °C and 5% CO$_2$. In the absence of CO$_2$-level maintenance, culture medium was buffered with 25 mM Hepes. Different imaging devices were used. Plasma membrane images were acquired on the same Nikon TIRF microscope as described above. Other live-cell images were acquired on a Zeiss Cell Observer Spinning Disk confocal device equipped with a Plan Apo 100× NA 1.4 oil immersion objective and an AxioCam MRm (Zeiss). Montages, kymographs, and movies were prepared with Fiji software (NIH). In addition, TIRF and spinning-disk images

(Supplementary Figs. 5d, 7e, 8a–c, 9g) were processed with the Fiji plugin "A trou Wavelet" (at default parameters) and then smoothed to improve contrast.

**Live-cell lattice light-sheet microscopy.** Genome-edited U2OS cells (CLTA-mRFP) were plated 20 h before imaging on 5-mm- diameter #1.5 coverglass. Cells were incubated for 1 min with anti-CD166-Atto647N antibody (5 μg ml$^{-1}$) or with transferrin-Atto647N (50 nM) at 21 °C in lattice light-sheet imaging media (phenol red-free DMEM, high glucose, glutamax, supplemented with 1% BSA, 0.01% penicillin–streptomycin, 1 mM pyruvate, and 20 mM HEPES (pH 7.3)). The coverslips were transferred to the lattice light-sheet microscope (LLSM) coverslip holder and inserted into the imaging chamber containing lattice light-sheet imaging media in the absence of $CO_2$ at 30 °C (for increased stability of the optical system). Imaging started within 2–3 min. Lattice light-sheet microscopy was performed on a commercialized instrument of 3i (Denver, USA), previously described[50]. Cells were scanned incrementally through a 20-μm-long light sheet in 600-nm steps using a fast piezoelectric flexure stage equivalent to ~325 nm with respect to the detection objective, and were imaged using a sCMOS camera (Orca-Flash 4.0; Hamamatsu, Bridgewater, NJ). Excitation was achieved with 560- or 642-nm diode lasers (MPB Communications) at 20% acousto-optic tunable filter transmittance with 100 mW initial box power through an excitation objective (Special Optics 28.6× 0.7 NA 3.74-mm water-dipping lens), and detected via a Nikon CFI Apo LWD 25× 1.1 NA water-dipping objective with a 2.5× tube lens. Lattice light-sheet imaging was performed using an excitation pattern of outer NA equal to 0.55 and inner NA equal to 0.5. Composite volumetric datasets were obtained using 20-ms exposure/slice/channel at a time resolution of 3 s per cell volume (60 slices). Fifty time points were acquired per cell. Acquired data were analyzed using an adapted version of cmeAnalysis3D software published previously[51]. It can be found as part of the Github repository of llsmtools in https://github.com/francois-a/llsmtools/. cmeAnalysis3D was implemented in Matlab 2018a. Quantification of LLS images was performed on raw images (see Supplementary Methods). Imaris 9.3 was used for the LLSM visualization and video rendering.

**Fluid-FM coupled to fast-scanning confocal microscope.** For the dynamic study of endoA3 localization to the site of galectin exposure, we used a combined instrument, made up of a Fluid-FM setup (Cytosurge) coupled to a Bruker Resolve AFM and to a Zeiss LSM880 fast-scanning confocal microscope equipped with a Plan Apo 63× numerical aperture (NA) 1.4 water immersion objective.

Twenty-four hours before the experiment, HeLa cells expressing endoA3-GFP were seeded on glass-bottom petri dishes. Just before the experiments, cells were incubated for 30 min at 37 °C in lactose buffer (20 mM Hepes, pH 7.2, 150 mM lactose, 45 mM NaCl, 5 mM KCl, 1 mM $MgCl_2$, and 1 mM $CaCl_2$) to strip galectins of the cell surface, and subsequently kept in serum-free culture media at 37 °C and 5% $CO_2$ for all the durations of the experiment.

In a typical live-cell experiment, a suspension of gold beads (10$^5$ nanoparticles ml$^{-1}$ in PBS, pH 7.4), functionalized with the fluorescently labeled Gal8 and Gal1 or with the control dye (fluorescent nanoparticle without protein coating), was placed on a glass-bottom petri dish and imaged with the confocal device. For experimental details on the coating of nanoparticles, see Supplementary Methods. A Fluid-FM microchanneled cantilever was approached to the surface of the petri and used to trap individual functionalized gold beads, by applying a steady negative pressure of −800 mbar through the microfluidics system. Using the force control of the Bruker Resolve atomic force microscope, the cantilever with the trapped functionalized nanoparticle was then approached to the membrane of individual endoA3-GFP-expressing HeLa cells by applying a 10 nN initial force that was subsequently decreased to 5 nN and kept constant over time. In parallel, the focal plane of the confocal device was focused on the bead, and 60-s time series were recorded in fast-scanning mode to monitor endoA3-GFP fluorescence variations in the surroundings of the bead. A control experiment in which the cells were approached by bare Fluid-FM probes (with no trapped nanoparticle) was also carried out.

**Flow cytometry.** For uptake assays by loss of surface (Figs. 1a, b, 2b; Supplementary Figs. 2a, b, 3a, d, 8g) and other cell surface analyses (Supplementary Fig. 4f, g), measurements were made with a Guava easyCyte (Merck-Millipore) flow cytometer (10,000 or 30,000 cells counted per condition). Flow cytometry was set up and gated as described in Supplementary Fig. 12. For the generation of the stable HeLa endoA3-GFP cell line, single-cell sorting in 96-well plates was performed with a FACSAria III (BD Biosciences).

**Knocksideways.** The knocksideways of α-adaptin was performed as described previously[52].

**Uptake assays.** Cells transfected with specified siRNAs or plasmid constructs were seeded in 4- or 24-well plates, 16–24 h before the experiment, in order to reach subconfluence the day of the experiment. In the following steps, culture medium was always buffered with 25 mM Hepes, and supplemented with 10% FBS for anti-CD166 uptake, but kept serum-free for transferrin, galectin, or β1-adrenergic receptor uptake.

**Loss of surface assays in flow cytometry.** In the case of anti-CD166 uptake, cells were pre-incubated for 30 min at 37 °C in fresh serum-containing culture medium just before the experiments, while this step was skipped for transferrin. Plates were then transferred on ice, and cells were washed with ice-cold culture medium. Anti-CD166 antibody (5 μg ml$^{-1}$) or biotinylated transferrin (Tf-Biot, 10 μg ml$^{-1}$) were bound to the cell surface on ice in culture medium for 30 min. After washes to remove unbound ligands, cells were put back to 37 °C in pre-warmed culture medium to induce endocytosis for various time lapses. One sample was kept on ice as 0-min time point, giving the total cell surface abundance of each cargo (or 0% endocytosis). Endocytosis was stopped by placing cells on ice and washing with ice-cold PBS$^{++}$ containing 0.2% BSA. Residual cell surface-accessible anti-CD166 antibody or Tf-Biot (corresponding to non-endocytosed fractions) were detected with a Alexa Fluor 647-labeled secondary antibody or streptavidin, respectively. After extensive washes with ice-cold PBS, cells were detached by incubation on ice in PBS containing 4 mM EDTA for 10–15 min. Cell suspensions were kept on ice until measurements by flow cytometry (10,000 cells for each condition). For both cargoes, the percentage of endocytosis at each time point was derived from the fraction of the remaining cell surface signal reported to the total surface signal at time 0. For gating strategies, see Supplementary Fig. 12.

**Continuous uptake in immunofluorescence.** For anti-CD166 antibody uptake alone, cells were pre-incubated for 30 min at 37 °C in fresh serum-containing medium just before experiments. In the particular case of uptake experiments under glycosphingolipid depletion, cells were pretreated for 2 days with 5 μM Genz-123346, and serum concentration was reduced to 5% to reduce salvation pathways in glycosphingolipid biosynthetic processes. For transferrin uptake alone or co-uptake with anti-CD166 antibody, cells were pre-incubated for 30 min at 37 °C in serum-free medium just before experiments. For co-uptake experiments of anti-CD166 antibody and fluorescently labeled Gal1, Gal3, or Gal8, cells were pre-incubated for 30 min at 37 °C in lactose buffer (20 mM Hepes, pH 7.2, 150 mM lactose, 45 mM NaCl, 5 mM KCl, 1 mM $MgCl_2$, and 1 mM $CaCl_2$) just before experiments to strip off galectins present in the cellular environment. Endocytosis was then triggered by incubating cells at 37 °C in the continuous presence of anti-CD166 antibody (5 μg ml$^{-1}$) and/or fluorescently labeled transferrin (Tf-A488, 10 μg ml$^{-1}$) or fluorescently labeled Gal1/Gal3/Gal8 (Gal1-Cy5, Gal3-A488, and Gal8-Cy3, 50 nM) in pre-warmed culture medium for various time lapses. Endocytosis was stopped on ice, and unbound ligands were removed by extensive washes with ice-cold PBS$^{++}$. An additional step of three 5-min washes on ice with lactose buffer was added in the experiments with Gal1/Gal8 to strip off residual galectin signal from cell surface. Residual cell surface-accessible anti-CD166 antibody or Tf-A488 were stripped by three 20-s acid washes on ice (200 mM acetic acid, pH 2.5, 300 mM NaCl, 5 mM NaCl, 1 mM $CaCl_2$, and 1 mM $MgCl_2$)[2]. After neutralization by extensive washes with ice-cold PBS$^{++}$, cells were fixed, permeabilized, incubated with primary and/or secondary antibodies if necessary, and mounted as previously described (see the "Light microscopy" section). Of note, for co-localization experiments with endoA isoforms, acid and lactose washes were skipped and replaced by a quick wash at 37 °C with pre-warmed PBS, and cells were subsequently fixed for 10–20 min at 37 °C in order to keep the integrity of endoA signal distribution. In addition, in the absence of a functional antibody against endoA3 in immunofluorescence, cells were transfected with an endoA3-GFP construct 16–24 h before experiments. Samples were imaged, and co-localization or internalized fluorescent signals was quantified as explained below. Of note, the efficiency of acid washes of anti-CD166 antibody was observed in immunofluorescence and quantified by flow cytometry, showing >95% removal of surface-bound antibody.

**Uptake after binding on ice in immunofluorescence.** For co-uptake of Gal3 and anti-CD166 antibody in HeLa and MEF cells, the binding of ligands to the cell surface was performed sequentially on ice for 20 min each in serum-free medium, the first one being Gal3 (Gal3-A488, 200 nM) and the second anti-CD166 antibody. This operation avoids the nonspecific binding of Gal3 to the glycosylations of antibodies. After binding of each ligand, cells were extensively washed with ice-cold PBS$^{++}$. Endocytosis was triggered by switching cells back to 37 °C with pre-warmed culture medium for 5–10 min. Endocytosis was stopped on ice, and unbound ligands were removed by extensive washes with ice-cold PBS$^{++}$. Lactose stripping was then performed as explained above. Subsequent removal of signal coming from cell surface-accessible anti-CD166 antibody was performed differently according to the cell line: acid stripping as above for HeLa; 45 min of blocking on ice with 100 μg ml$^{-1}$ unlabeled secondary antibodies for MEF. After extensive washes with ice-cold PBS$^{++}$, cells were fixed, permeabilized, incubated with fluorescent secondary antibody, and mounted as previously described (see the "Light microscopy" section). Samples were imaged, and co-localization was quantified as explained below.

**Uptake in live-cell imaging.** Cells transiently or stably expressing endoA3-GFP were maintained in serum-containing culture medium at 37 °C under spinning-disk or TIRF microscope. Fluorescently labeled anti-CD166 antibody (5 μg ml$^{-1}$) was added continuously in the culture medium, and cells were imaged for maximum 2 min up to 10 min after antibody addition.

**Rescue of CD166 uptake by Gal8**. For this experiment, cells were previously transfected with AllStars Negative siRNA or siRNAs against endogenous Gal8. Just before the experiment, cells were pre-incubated for 30 min at 37 °C in lactose buffer (20 mM Hepes, pH 7.2, 150 mM lactose, 45 mM NaCl, 5 mM KCl, 1 mM MgCl$_2$, and 1 mM CaCl$_2$) to strip off galectins present in the cellular environment. Cells were then incubated for 10 min at 37 °C in the continuous presence of anti-CD166 antibody (5 μg ml$^{-1}$) in pre-warmed serum-containing or serum-free medium, or serum-free medium supplemented with increasing doses of exogenous Gal8 (0.2–20 nM). Endocytosis was stopped on ice, and unbound ligands were removed by extensive washes with ice-cold PBS$^{++}$. Residual cell surface-accessible galectins were stripped by three 5-min washes on ice with lactose buffer (see above). Residual cell surface-accessible anti-CD166 antibody was stripped by three 20-s acid washes on ice (see above). After neutralization by extensive washes with ice-cold PBS$^{++}$, cells were fixed, permeabilized, incubated with secondary antibody, and mounted for observation as previously described (see the "Light microscopy" section).

**β1-adrenergic receptor uptake**. β1-AR stimulation was performed by incubating BSC-1 cells at 37 °C for 4 min with pre-warmed serum-free culture medium containing 10 μM dobutamine. After incubation, cells were quickly washed once at 37 °C with pre-warmed PBS to remove unbound ligand, and fixed with pre-warmed 4% paraformaldehyde for 20 min at 37 °C. Afterward, fixed cells were permeabilized and immunostained for β1-AR and endoA2. For endoA3 detection, cells were transfected with an endoA3-GFP construct 16–24 h before the experiment. Resting conditions correspond to cells being cultured in 10% serum medium and directly fixed. Of note, cells were never serum-starved or incubated at 4 °C before experiments.

**Pull-down assays**. In all, 2 L cultures of BL21 *Escherichia coli* transformed with GST-CD166 tail and GST constructs (see above) were incubated for 3 h at 37 °C with 100 μM IPTG for induction. After centrifugation, cell pellets were resuspended in a solution of 150 mM NaCl, 25 mM HEPES, pH 7.4, and 2 mM EDTA complemented with 1 mM PMSF, 2 mM DTT, 100 μg ml$^{-1}$ lysozyme, and protease inhibitor cocktail. After incubation on ice for 30 min, cells were lyzed by snap-freezing in liquid nitrogen, quick thawing, and brief sonication. Lysates were clarified by ultracentrifugation in a Beckman Ti45 rotor for 1 h at 41,000 rpm (~131,000 × *g*). Following centrifugation, the supernatant was bound to 1-ml column bed of glutathione beads (GE Healthcare). The beads were washed 5 times with 10 column volumes of wash buffer (500 mM NaCl, 25 mM HEPES, pH 7.4, and 2 mM EDTA). Finally, protein-bound beads were resuspended in 1 ml of lysis solution (150 mM NaCl, 25 mM HEPES, pH 7.4, and 2 mM EDTA).

HeLa cells stably expressing GFP-FKBP-tagged endoA2 and endoA3 were quickly washed with ice-cold PBS, lysed in ice-cold lysis buffer (150 mM NaCl, 25 mM HEPES, pH 7.4, and 2 mM EDTA complemented with 1 mM PMSF, 1% NP40, protease, and phosphatase inhibitor cocktail (PhosSTOP roche)), sonicated, and spun at 14,000 × *g* for 15 min. In all, 100 μl of GST/GST-CD166 tail-bound beads were then exposed to cell lysates for 2 h at 4 °C, pelleted in a cooled benchtop centrifuge, and washed five times in lysis buffer complemented with 1% NP40 only. Beads were then boiled for 10 min in 100 μl of sample buffer for elution, and 8 μl of sample were run on sodium dodecyl sulfate polyacrylamide gel electrophoresis, ("input" lanes correspond to 1.5% of cell extract (final concentration of ~0.04 mg ml$^{-1}$)). The proteins were transferred and immunoblotted using anti-endoA2, anti-endoA3, and anti-ezrin antibodies. Finally, membranes were stained with Coomassie blue to control the amount of GST fragments in each sample.

**STxB internalization and tubulation**. Transport of STxB to perinuclear Golgi, or observation of plasma membrane STxB-induced tubules, were performed as previously published[4]. For further details, see Supplementary Methods.

**Wound-healing assay**. In total, 25,000–55,000 cells previously treated for 48 h with siRNAs were seeded in each compartment of wound-healing assay silicone inserts (Ibidi, 80209) placed in the wells of a glass-bottom 24-well SensoPlate dish (Greiner, 662892). Cells were grown for an additional 16–24 h to reach confluency. Silicone inserts were removed, cells were washed with PBS, and fresh culture medium was added. Cell movements were followed for up to 48 h at 37 °C and 5% CO$_2$ in bright-field illumination using a Zeiss LSM 710 confocal microscope equipped with an EC Plan-Neofluar 10× numerical aperture (NA) 0.3 dry objective and a cabinet for control of temperature and CO$_2$. Images were acquired every 10 min at the same position for each sample, using time lapse and multiposition acquisition modules of Zen 2.3 sp1 blue edition software.

**Cell–cell adhesion flipping assay**. Cell–cell adhesion flipping assays were performed as previously described[44,45]. Briefly, a suspension of fluorescently labeled cells in PBS$^{++}$ was sedimented for 10 min in a chamber slide containing unlabeled confluent monolayers of adherent cells. A picture was taken before flipping the dish. Slides were then filled and carefully dipped in a large vessel containing pre-warmed PBS$^{++}$. They were then rotated by 180° and maintained in an upside-

down position for 15 min, allowing cells that did not adhere to the monolayer to detach. The chamber slide was then rotated back 180° and carefully removed from the large vessel. A second picture was taken after flipping, around the same position. Finally, the percentage of cells in suspension adhering to the monolayer of adherent cells was calculated. For this assay, wide-field images were rapidly acquired on a Zeiss Axio Observer 7 epifluorescence microscope equipped with a EC Plan-Neofluar 5× NA 0.16 dry objective and an ORCA-Flash 4.0 LT sCMOS camera (Hamamatsu).

**Image quantifications**. All image quantifications were performed with ImageJ/Fiji v2.0.0-rc-65/1.51u (NIH) and Icy v1.9.10.0 (Institut Pasteur) softwares, except for LLSM images. For LLSM image quantifications, please see the specific paragraph.

**Quantification of cargo uptake from confocal images**. For further details, see Supplementary Methods.

**Quantification of STxB transport to Golgi apparatus**. For further details, see Supplementary Methods.

**Quantification of co-localization**. Quantification of co-localization between two channels was performed using tools previously described[4]. For further details, see Supplementary Methods.

**Quantification of spot lifetime from TIRF images**. For further details, see Supplementary Methods.

**Quantitative analysis of live-cell lattice light-sheet images**. For further details, see Supplementary Methods.

**Quantification of endoA3 recruitment to plasma membrane**. For further details, see Supplementary Methods.

**Quantification of gap closure in wound-healing assays**. For further details, see Supplementary Methods.

**Single-cell tracking in wound-healing images**. For further details, see Supplementary Methods.

**Quantification of cell–cell adhesion in flipping assays**. For further details, see Supplementary Methods.

**Statistical analyses**. All statistical analyses were performed using Prism v8.3.0 software (Graphpad Inc). The normality of datasets was checked with D'Agostino–Pearson omnibus normality test. In the case of Gaussian distributions, parametric tests were used, and data were represented on graphs as mean ± s.e.m as error bars. In the case of non-Gaussian distributions, nonparametric tests were used, and data were represented on graphs as median ± 95% CI as error bars. Details on the parametric and nonparametric tests used for each analysis, as well as other statistical details related to specific graphs, are indicated in figure legends. Significance of comparisons is represented on the graphs by exact *p* values and/or by asterisks. Exact *p* values are shown only when provided by the statistical software. No statistical method was used to predetermine sample size.

**Reporting summary**. Further information on research design is available in the Nature Research Reporting Summary linked to this article.

## Data availability

The authors declare that the main data supporting the findings of this study are available within the paper and its Supplementary Information files. The source data underlying each graph in Figures and Supplementary Figures are provided as a Source Data file. The mass spectrometry proteomics data generated during this study, and used for the analysis presented in Supplementary Data 1 and Supplementary Fig. 1a, have been deposited to the ProteomeXchange Consortium via the PRIDE partner repository with the dataset identifier PXD017526 [http://proteomecentral.proteomexchange.org/cgi/GetDataset?ID=PXD017526]. The sequences of human endoA proteins used in this study are available in Swissprot database with the accession codes Q99962 [https://www.uniprot.org/uniprot/Q99962] (for endoA1), Q99961 [https://www.uniprot.org/uniprot/Q99961] (for endoA2), and Q99963 [https://www.uniprot.org/uniprot/Q99963] (for endoA3). Extra data are available from the corresponding authors on reasonable request.

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

## Acknowledgements

We would like to acknowledge the following people for help in experiments and providing materials or expertise: P. de Camilli, V. Fraisier, T. Kirchhausen, L. Leconte, H.T. McMahon, C. Merrifield, M. S. Robinson, L. Sengmanivong, and E. Smythe. We thank B. Knoops, P. Dumont, and F. Gofflot from LIBST (UCLouvain) for sharing cell culture and flow cytometry facilities and materials. We greatly acknowledge IMABIOL imaging platform from LIBST (UCLouvain) and M.-C. Eloy for technical assistance; PICT imaging platform from de Duve Institute (UCLouvain); P. van der Smissen for technical assistance. We thank A.-M. Faber and H. Degand for their help in sample preparation and preliminary proteomic analysis. We thank D. Brusa and the Flow Cytometry platform of IREC (UCLouvain), and of the Institut Curie for their contribution. We also thank P. Gilon and H.-Y. Chae from IREC (UCLouvain) for giving access to their TIRF microscope and technical help, that allowed us to do preliminary observations. We are very grateful to V. Chambon for the production of fluorescently labeled STxB. We thank Abberior Instruments GmbH for the acquisition of STED images of our samples on their demonstration device. This work was supported by grants from the "Fonds National de la Recherche Scientifique" (FNRS, CDR-J.0119.19) and the "Communauté française de Belgique–Actions de Recherches Concertées" (17/22-085). This work was also supported by the French National Research Agency (DALLISH–ANR-16-CE23-0005), and by Inria in the frame of NAVISCOPE-IPL (Inria Project Lab). The bioprofiling platform used for the proteomic analysis was supported by the FNRS, the European Regional Development

Fund, and the Walloon Region, Belgium. The LLSM was financed by PIA France-Bioimaging (ANR-10-INBS-04_01), LabEx DCBiol, LabEx CelTisPhyBio ANR-11-LABX-0038, Agence Nationale de la Recherche (ANR-16-CE23-0005-02, ANR-19-CE13-0001-01), HFSP (RGP0029/2014), and European Research Council (ERC project 340485). We greatly acknowledge the Cell and Tissue Imaging Facility (PICT-IBiSA) and Nikon Imaging Centre, Institut Curie, member of the French National Research Infra-structure France-BioImaging (ANR-10-INBS-04). H.-F.R. is a FNRS postdoctoral research fellow (Belgium). F.T., T.H. and C.L. are supported by PhD fellowships from FRIA/FNRS (Belgium). C.L.G. is an EMBO Long-term postdoctoral fellow. P.V.D.B. and D.A. are supported by WELBIO (Fédération Wallonie-Bruxelles, Belgium). This work was supported by the European Research Council under the European Union's Horizon 2020 research and innovation program (grant agreement no. 758224). D.A. is a research associate of the FNRS (Belgium).

## Author contributions

The following datasets were contributed by the indicated authors: Figs. 1a–d, 2a, b, e, 3a, b, 4a–g, and Supplementary Figs. 1, 2a, b, 3a, b, d, f, 4a–f, h, 5b, d–g, 7a, b, e, 8a–g, 9a, c–e, g, h, 10–12 (HFR); Figs. 2c, d, f–h, 3c–f, 4f, h, and Supplementary Figs. 2c–h, 3c, e, 4g, 5a, c, 6, 7c, d, 8h, 9f, i–l, 11, 12 (FT); Fig. 3d–f, and Supplementary Fig. 9i–l (C.L.G.); Supplementary Figs. 3e, 10d, e (T.H.); Fig. 1e–h, and Supplementary Fig. 2i (C.A.V.C. and C.W.); Supplementary Fig. 4h (C.L.); Supplementary Fig. 9b (M.S.Z.); Supplementary Fig. 1a (R.W.). CW purified galectins. L.J., M.S.Z., and R.W. provided technical support and conceptual advice. H.F.R., P.M., D.A., P.V.D.B., and C.W. provided direction and guidance. H.F.R. and P.M. conceived the initial design of the study. H.F.R. wrote the first draft. All authors discussed the results and commented on the paper.

## Competing interests

The authors declare no competing interests.
