## [Peer Review File · Nature Communications]

Reviewers' comments:

Reviewer #1 (Remarks to the Author):

How clathrin-independent endocytosis (CIE) is regulated and its biological relevance to cancer are important questions. In this study, the authors identified by iTraQ the tumor marker CD166 as a CIE cargo and found that enophilin A3 is required for its internalization. The authors also provide evidence that endo A3 mediates a CIE pathway that is distinct from endo A2-dependent ones. Further, they show by Fluid-FM that Galectin-8 drives CIE of CD166 by recruiting endo A3 to uptake sites at plasma membrane. Finally, the authors show by wound healing experiment that endo A3 is required for cancer cell migration. They also demonstrate that proliferation of CTL relies on the surface levels of CD166 of target cells. Based on these findings the authors propose that downregulation of CD166 from cell surface by endo A3-dependent CIE promotes cancer cell migration and decreases their immunogenicity, which might contribute to tumor metastasis.

The identification of the role of endo A3 as new player in CIE is novel and important finding for understanding mechanisms for clathrin-independent endocytosis. The authors applied cutting-edge imaging technology to provide convincing evidence that CD166 is internalized via clathrin-independent, Gal8- and endo A3-dependent mechanism. The experiments are carefully carried out and well controlled. I would also like to compliment the authors for providing very detailed experimental protocols in Materials and Methods. However, the last part of the study is less convincing due to lack of direct evidence for the pathophysiological role of endo A3-mediated CIE of CD166 in cancer because the LB33-MEL melanoma cells are endo A3 negative. To conclude that endo A3 regulates the immunogenicity of cancer cells by downmodulation of surface levels of CD166, the authors should either overexpress endo A3 in the melanoma cells or knockdown its expression in HeLa cells stably expressing the HLA molecules and determine their immunogenicity by CTL proliferation assay.

I have otherwise only minor concerns about this manuscript. When the major point mentioned above and minor issues as listed below have been addressed, I would recommend publication in Nature Communications.

Minor points:

1. Line 150-153: Figure 2d shows the quantitative data of the effects of endoA knockdown on STxB transport. It would be more convincing to include confocal images in the figure to show the subcellular distribution of STxB in siRNA-transfected cells.
2. Line 235-236: the authors stated that "Strikingly, this effect of endoA3 depletion vanished in the absence of CD166". However, in Figure 4b and Movie 13, knockdown of CD166 did not fully restore the gap closure speed of endoA3-depleted cells to the levels of control cells. The authors should modify this statement to more accurately interpret the data.

Reviewer #2 (Remarks to the Author):

Renard et al. describe a novel mechanism of clathrin-independent endocytosis (CIE) responsible for the internalization of CD166/ALCAM. This mechanism resembles similar characteristics to the CLIC/GEEK pathway responsible for internalization of Shiga toxin and some cellular cargoes like CD44. Differently from that mechanism, the one described in this manuscript requires the specific action of Endophilin A3 (but not of Endophilin A2 or A1) and Galectin-8 (but not Galectin1 or 3). These data suggest that specific mechanisms of CIE are involved in the internalization of specific cargoes with peculiar molecular requirements. The pathway described here seems to have a role in the downmodulation of cell surface levels of CD166/ALCAM with potential effects on cell migration and immunogenic response.

The data presented are of potential for the community of membrane trafficking and of cell biology in general, as they shed light on the heterogeneity of CIE pathways, their cargo specificity and their potential relevance for cell physiology. However, the data remain mainly at the phenomenological level, with no mechanistic studies. For instance, what are the molecular basis for Endophilin A3 specific role vs. the other members of the family? Is Endophilin A3 directly recruited to the CD166 cargo or it is recruited to PM microdomains enriched in CD166 and sphingolipids (as compared to the other family members)? Similarly, for Galectin-8 there are no further investigations about mechanism and specificity. Finally, the data on the possible impact on migration and immunogenic response are very interesting, but still very preliminary. For these reasons, I think that the manuscript, in its current state, does not reach sufficient level of mechanistic insights required for publication Nature Communication. In addition to the general comment, there are also a number of more specific issues listed below.

- 1) Authors started from a screening looking for proteins whose level was decreased upon clathrin-dependent endocytosis inhibition, claiming that this might be the consequence of upregulation of a CIE pathway leading to degradation. With this argument in mind, they selected CD166 for their study as it was downregulated upon inhibition of clathrin-dependent endocytosis both at protein and mRNA level. However, in contrast with their initial hypothesis, CD166 internalization is not accelerated by clathrin-KD, AP2-KD or upon other inhibitors, suggesting that CIE is not upregulated. Thus, the issue remains open (and not discussed in the manuscript): why CD166 protein levels are downregulated upon clathrin-mediated endocytosis inhibition? Is this reflecting the transcriptional effect and/or clathrin/AP2 has an impact on CD166 later trafficking steps (not on CIE)? More investigations are needed to clarify this point.
- 2) The data on dynamin is not convincing. There's a clear effect at 5 and 10 min (more than 50% magnitude of inhibition), exactly when early steps of internalization are measured. I understand that this is not statistically significant, but more statistics is needed to exclude a role.
- 3) The physiological impact of the pathway on CD166-dependent migration and immunogenic response is preliminary. Authors should provide stronger molecular genetic evidences for a causal link with Endophilin A3/Galectin-8 mechanism.
- 4) Minor point: note that Endophilin A1, A2 and A3 are not "isoforms" as they are products of different genes. The nomenclature along the manuscript is wrong and need to be revised.

Reviewer #3 (Remarks to the Author):

In this manuscript, the authors identify novel cargos of the Non-Clathrin-dependent endocytic (NCE) pathways. They, in particular, focus on CD166 (ALCAM) and provide a set of convincing evidence that this cargo enters an NCE route that utilizes endophilins as membrane curvature, BAR-containing determinant. Importantly, they specifically show that Endophilin-A3 mediates the internalization of CD166 in various cell types. Collectively, these results support the existence of cargo and adaptor-specific entry routes in NCE. This study, in particular, challenges the prevailing view that the three endophilin-A isoforms are functionally redundant in NCE. This is interesting. However, while the data presented do support this contention and further enable the authors to identify a specific cargo, CD166, that uses EndoA3, there is little effort to understand where the specificity comes from and neither suggestions nor experiments as to the possible mechanisms underlying the specificity of the different Endophilins for different cargos, different vesicles and distribution pattern. A number of key questions remain unaddressed. For example, does EndoA3 interact with CD166? Can a structure function analysis of the 3 endophilins reveal insights into how they might act and in an ostensibly specific fashion? These and a number of issues underlined below weaken the enthusiasm for this manuscript and question whether the findings reported are sufficient to justify publication in NC.

Specific points

1. The authors start by a mass spectrometry approach to identify cell surface proteins that are

altered upon inhibition of Clathrin-mediated endocytosis (CME), and focus specifically on CD166, the levels of which are severely reduced after Clathrin ablation. This data is interesting but the logical premises would need to be clarified. Specifically, it is unclear as to why the authors decided to focus on proteins that display a reduced cell surface levels after inhibition of Clathrin in order to identify non-Clathrin endocytic cargos. In the most obvious scenario, perturbation of CME should not affect the entry of cargo that use a different entry route, yet they end up studying CD166 whose cell surface levels are severely reduced following Clathrin inhibition.

2. As comment to figure 1d, it is stated that: "In addition, continuous co-incubation of cells with fluorescently labeled anti-CD166 antibody and transferrin clearly showed that both cargoes do not co-localize (Figure 1d)." While the data do support a lack of colocalization between these two cargos, their interpretation as indication of the fact the cargos under scrutiny enter via distinct routes is unwarranted. Indeed, these experimental results are surprising as one would expect that after entry into cells some levels of colocalization between NCE and CME cargos in either early or late endosome should be detected. The lack of such colocalization does require some explanation and should not be used as indication that CD166 employs an entry route different from CME. A technical, but relevant issue to be addressed in this context is whether after continuous incubation with anti-CD166 and anti-TFR the surface bound abs have been completely removed (by acid washes) to ensure that only internalized abs are followed. In the methods section, 3 rapid (20") acid washes are described but it is unclear whether experimentally the authors verified that after this treatment CD166 antibodies had been removed from the cell surface.

3. In Figure 2A and Suppl-3A, it is shown that ablation of Endophilin A3, but not of A1 or A2 impairs the internalization of CD166. However, in Fig. 2A what is measured is the relative amount of internalized CD166 with respect to mock transfected cells, whereas in Suppl-3A the relative levels of cell surface CD166 are reported. This second measurement indicates that inhibition of Endo-A3 only marginally reduces CD166 surface levels (from ~ 40% to ~ 30%), suggesting Endo-A3 is, at best, partially responsible for CD166 entry and other routes must be at play. Hence, caution in interpreting these results should be used. These findings also beg the question as to whether the concomitant inhibition of Clathrin and EndoA3 impacts more dramatically on CD166 surface levels and endocytosis.

4. The Plot in figure 2b that correlates the levels of endocytosis with Endo A1, A2 and A3 expression is redundant and add little information to the data reported in 2A.

5. "In addition, overexpression of endo-A3 – but not endo-A2 – significantly increases the uptake of CD166 (Extended Data Figure 3c)". It is unclear how to interpret these results. At face value, they would suggest that Endo-A3 is limiting in determining the entry of CD166. However, to corroborate this conjecture it would seem necessary to restore the expression of EndoA3 in those cells where its levels are low or undetectable, such as SUM159, LB33-MEL and MZ2-MEL.43 cells, and test CD166 internalization.

6. "Interestingly, TIRF imaging of living cells showed that GFP-labeled endoA3 has a very short life-time at cell surface, as ~90 % of spots lasted for less than 30 seconds (Extended Data Figure 5e-g, green bars on the histogram). Interestingly, endoA3 became much less dynamic at sites of CD166 clustering, as ~80 % of endoA3 spots had a life-time exceeding 50 seconds and often the total duration of our 120 sec movies (Extended Data Figure 5e-g, red bars on the histogram; Supplementary material, Movie 6)"

Here, the authors used TIRF to study the dynamics of CD166 and EGFP-EndoA3. It would be good to use similar TIRF-based analysis to study CD166 dynamics in the absence of Endo-A3 and upon its overexpression. Indeed, the authors points that at sites of EndoA3 clustering the dynamics of CD166 is robustly slowed down. This is apparently in contrast with the findings reported above that elevation of EndoA3 increases CD166 internalization and some explanation is required.

7. The authors provide evidence that EndoA1, A2 and A3 display distinct cellular localization patterns. The data are overall convincing but their outcome could be strongly influenced by the sensitivity of the antibody used or the levels of expression of EndoA3-GFP. Some cautionary wording would be welcomed. Alternatively, an EM study of the various Endophilin isoforms might address whether they are indeed distinctly localized.

8. 10 mM LacNAc treatment is shown to inhibit robustly CD166 entry, to a significantly larger extent with respect to Endo-A3 ablation. This might be due to a number of differences in the two

experimental conditions. Nevertheless, one would expect that after Endo-A3 inhibition, when a sizable fraction of CD166 still enters the cells as judged by the data reported in S3A, the addition of 10 mM LacNAc should not further reduce CD166 entry if, as proposed, the Endo-A3 endocytic route is galectin-dependent. Similar experiments should also be performed using siRNA against Galectin 8

9. In figure 4, the authors attempt to broaden the meaning of EndoA3-dependent NCE exploring the impact on tumoral cell migration and T cell proliferation. They initiate by showing that EndoA3 siRNA impairs wound closure of U2OS, only when these cells express CD166. This is an elegant experiment that, however, would need to be strengthened by the use of additional cancer cells, and specifically of tumor cells that do not express EndoA3, but still possess CD166. It would also be necessary to test CD166 internalization in U2OS and to test the impact of EndoA3 ablation on wound healing in HeLa cells.

In general, the attempts to link Endo-A3 to cancer biology appears weak, at best preliminary precluding the authors to draw any meaningful conclusions as to the role of Endo-A3 in cancer biology, and specifically on cancer cell migration/ invasion.

Manuscript NCOMMS-19-30071-T: Answers to reviewers

We would like to thank the three reviewers for their appreciation of our work and their constructive input, that allowed us to improve substantially our manuscript.

In order to make the reviewing work easier, we highlighted the main modified parts of the manuscript in blue color and added line numbers. Of note, we improved the title of the manuscript, so that it better reflects the content of our story.

Reviewers' comments:

Reviewer #1 (Remarks to the Author):

How clathrin-independent endocytosis (CIE) is regulated and its biological relevance to cancer are important questions. In this study, the authors identified by iTraQ the tumor marker CD166 as a CIE cargo and found that enophilin A3 is required for its internalization. The authors also provide evidence that endo A3 mediates a CIE pathway that is distinct from endo A2-dependent ones. Further, they show by Fluid-FM that Galectin-8 drives CIE of CD166 by recruiting endo A3 to uptake sites at plasma membrane. Finally, the authors show by wound healing experiment that endo A3 is required for cancer cell migration. They also demonstrate that proliferation of CTL relies on the surface levels of CD166 of target cells. Based on these findings the authors propose that downregulation of CD166 from cell surface by endo A3-dependent CIE promotes cancer cell migration and decreases their immunogenicity, which might contribute to tumor metastasis.

The identification of the role of endo A3 as new player in CIE is novel and important finding for understanding mechanisms for clathrin-independent endocytosis. The authors applied cutting-edge imaging technology to provide convincing evidence that CD166 is internalized via clathrin-independent, Gal8- and endo A3-dependent mechanism. The experiments are carefully carried out and well controlled. I would also like to compliment the authors for providing very detailed experimental protocols in Materials and Methods. However, the last part of the study is less convincing due to lack of direct evidence for the pathophysiological role of endo A3-mediated CIE of CD166 in cancer because the LB33-MEL melanoma cells are endo A3 negative. To conclude that endo A3 regulates the immunogenicity of cancer cells by downmodulation of surface levels of CD166, the authors should either overexpress endo A3 in the melanoma cells or knockdown its expression in HeLa cells stably expressing the HLA molecules and determine their immunogenicity by CTL proliferation assay.

Answer of authors:

We thank the reviewer #1 for his/her appreciation of our work and his/her support. We totally agree with the reviewer that the last part of our study regarding the role of endoA3-mediated CIE of CD166 on immunogenicity of cancer cells was still preliminary. Unfortunately, in our experimental conditions, we could not observe any significant change in the proliferation of T cells when endoA3 is depleted from cancer cells. We think that going deeper into the role of endoA3-mediated CIE of CD166 on immunogenicity of cancer cells will require to investigate new readouts, and that goes beyond the scope of the present study. In order to preserve the clarity and maintain the homogeneity of the manuscript, we have

decided to remove this preliminary immunological data, in agreement with the editor. However, we added new data that strengthen the role of endoA3-mediated CIE of CD166 in cancer cell adhesion and migration, and we hope the reviewer will appreciate them (please, see point C of our answer to the general comments of reviewer #2).

I have otherwise only minor concerns about this manuscript. When the major point mentioned above and minor issues as listed below have been addressed, I would recommend publication in Nature Communications.

Minor points:

1. Line 150-153: Figure 2d shows the quantitative data of the effects of endoA knockdown on STxB transport. It would be more convincing to include confocal images in the figure to show the subcellular distribution of STxB in siRNA-transfected cells.

Answer of authors:

We thank the reviewer for this remark. We added representative images, showing the variations of STxB distribution for the different endoA depletion conditions (Extended Data Figure 7a), and modified the text of legends of Figure 2e (page 9, line 413) and Extended Data Figure 7a (page 23, lines 733-736) accordingly.

2. Line 235-236: the authors stated that “Strikingly, this effect of endoA3 depletion vanished in the absence of CD166”. However, in Figure 4b and Movie 13, knockdown of CD166 did not fully restore the gap closure speed of endoA3-depleted cells to the levels of control cells. The authors should modify this statement to more accurately interpret the data.

Answer of authors:

We agree with the reviewer that this sentence may be misleading. We modified the text accordingly (page 5, lines 257-260): "Strikingly, in the absence of CD166, depletion of endoA3 did not have any additional effect on gap closure (Figure 4a,b; Supplementary material, Movie 14). This result indicates that the migratory phenotype caused by the absence of endoA3 is tightly dependent on the presence of CD166."

Reviewer #2 (Remarks to the Author):

Renard et al. describe a novel mechanism of clathrin-independent endocytosis (CIE) responsible for the internalization of CD166/ALCAM. This mechanism resembles similar characteristics to the CLIC/GEEK pathway responsible for internalization of Shiga toxin and some cellular cargoes like CD44. Differently from that mechanism, the one described in this manuscript requires the specific action of Endophilin A3 (but not of Endophilin A2 or A1) and Galectin-8 (but not Galectin1 or 3). These data suggest that specific mechanisms of CIE are involved in the internalization of specific cargoes with peculiar molecular requirements. The pathway described here seems to have a role in the downmodulation of cell surface levels of CD166/ALCAM with potential effects on cell migration and immunogenic response.

The data presented are of potential interest for the community of membrane trafficking and of cell biology in general, as they shed light on the heterogeneity of CIE pathways, their cargo specificity and their potential relevance for cell physiology. However, the data remain mainly at the phenomenological level, with no mechanistic studies. For instance, what are the molecular basis for Endophilin A3 specific role vs. the other members of the family? Is Endophilin A3 directly recruited to the CD166 cargo or it is recruited to PM microdomains enriched in CD166 and sphingolipids (as compared to the other family members)? Similarly, for Galectin-8 there are no further investigations about mechanism and specificity. Finally, the data on the possible impact on migration and immunogenic response are very interesting, but still very preliminary.

For these reasons, I think that the manuscript, in its current state, does not reach sufficient level of mechanistic insights required for publication in Nature Communications.

In addition to the general comment, there are also a number of more specific issues listed below.

Answer of authors:

We thank the reviewer #2 for his/her positive opinion on our findings and his/her constructive input, that undoubtedly helped us to improve our manuscript.

A) Mechanism and specificity of endoA3 function in CD166 endocytosis:

To address his/her main concern regarding mechanistic aspects, we performed pull-down experiments from HeLa cell lysates using the cytosolic fragment of CD166 (fused to GST, produced in bacteria and bound to glutathione resin). Interestingly, we could pull down endoA3, but not endoA2. Of note, we could also pull down ezrin, used here as a positive control. Interaction of ezrin with the cytosolic tail of CD166 has previously been documented (Tudor, C., et al. 2014. Syntenin-1 and Ezrin Proteins Link Activated Leukocyte Cell Adhesion Molecule to the Actin Cytoskeleton. J Biol Chem. 289(19):13445-13460). This physical interaction of endoA3 with CD166 may explain the specificity of this new endocytic modality. These data were added in the manuscript in Figure 2d and the main text was modified accordingly (page 3, lines 150-157). Of course, we cannot exclude the role of specific microdomains/lipids in recruitment mechanism of endoA3. However, this question is out of the scope of the present manuscript and will be addressed in future studies. Nevertheless, a sentence regarding this aspect was added in the discussion (page 7, lines 327-330).

B) Mechanism and specificity of Gal8 function in CD166 endocytosis:

In addition, the reviewer questioned the mechanism and specificity of Gal8 in our process. Uptake experiments (Figure 3a,c and Extended Data Figure 9b,c,e) as well as Fluid-FM data (Figure 3e,f and Extended Data Figure 9i-k) already included in the first version of the manuscript brought a significant answer to this question: this endocytic modality seems specific to Gal8, but not Gal1 nor Gal3. In this revised version of the manuscript, we added new Fluid-FM data showing that Gal8 does not induce the recruitment of endoA2 isoform (Figure 3f; Extended Data Figure 9l). The text was modified accordingly (page 5, lines 236-238). Altogether, our data suggest that the mechanism described here is specific to Gal8 and endoA3.

Regarding mechanistic aspects, we already proposed in the previous version of the manuscript that the endocytic modality we discovered may operate according to the GL-Lect hypothesis. To verify this possibility, we performed new experiments where we depleted glycosphingolipids from cells using Genz-123346 drug (HY-12744A). After 2 days of acute treatment with this inhibitor of glycosphingolipid synthesis, we could observe a significant 50% reduction of anti-CD166 uptake (Extended Data Figure 9b, top right panel), which is similar to the reduction we observed in the presence of LacNac (Extended Data Figure 9a) or upon depletion of Galectin-8 (Figure 3b). Of note, Gal3 and Tf uptakes were used as controls of GL-Lect and non-GL-Lect cargoes, respectively (Extended Data Figure 9b, bottom left and bottom right panels). Efficiency of Genz treatment was also verified by the loss of STxB binding to cell surface (Extended Data Figure 9b, top left panel). The text was modified accordingly (page 4, lines 199-205). Together, these data demonstrate that the endocytic mechanism of CD166 is part of the GL-Lect hypothesis, as it is dependent on Galectins (Gal8 in particular) and on glycosphingolipids.

C) Impact of endoA3-dependent CIE of CD166 on cancer cell migration and immunogenicity:

Regarding migration and immunogenic response, we agree with the reviewer that data presented in the previous version of the manuscript were still preliminary. Unfortunately, in our experimental conditions, we could not observe any significant change in the proliferation of T cells when endoA3 is depleted from cancer cells. We think that going deeper into the role of endoA3-mediated CIE of CD166 on immunogenicity of cancer cells will require to investigate new readouts, and that goes beyond the scope of the present study. In order to preserve the clarity and maintain the homogeneity of our manuscript, we have decided to remove this preliminary immunological data, in agreement with the editor. However, we added new data that strengthen the role of endoA3-mediated CIE of CD166 in cancer cell adhesion and migration, and we hope the reviewer will appreciate them.

First, we added single cell tracking data obtained from the wound healing assays on U2OS cells (Figure 4c-e; Extended Data Figure 10a). We can clearly observe that endoA3 depletion strongly reduces the migration velocity of individual cells at the edge of the wound. Interestingly, the depletion of CD166 had little impact on the migration velocity (Figure 4d). Furthermore, in the context where CD166 is absent, additional depletion of endoA3 did not further affect cell migration velocity. This data further underlines that the phenotype of

endoA3 depletion requires the presence of CD166. Of note, in all conditions, the directionality of migration towards the wound (measured with xFMI, new Figure 4e) and the Euclidian distance (new Extended Data Figure 10a) are significantly decreased in comparison to the control condition.

In the absence of *endoA3* (where cells dramatically lose migration velocity and directionality, which in turn strongly reduces gap closure; see Figure 4a,b), we observed that CD166 accumulates at the cell surface (Extended Data Figure 4f,g). Interestingly, we could show in both HeLa and U2OS cell lines that this accumulation happens at intercellular contacts (Figure 4f). This led us to hypothesize that it could increase intercellular adhesion. In order to address this hypothesis, we tested cell-cell adhesion using previously described flipping assays (Langhe, R.P., et al. 2016. Cadherin-11 localizes to focal adhesions and promotes cell-substrate adhesion. *Nat Comm.* 7:10909; Kashef, J., Franz, C.M. 2015. Quantitative methods for analyzing cell-cell adhesion in development. *Dev Biol.* 401(1): 165-174; see Extended Data Figure 10h and methods section for more information). Briefly, this assay consists in looking at the percentage of cells added in suspension that adhere to a confluent monolayer of adherent cells. First, we observed the adhesiveness of cells treated with negative control siRNA, or siRNAs against CD166, *endoA3* or the combination, to a wild-type (untreated) confluent cell layer (Extended Data Figure 10i). This experiment clearly showed that a lack of CD166 significantly reduces intercellular interactions. In a second experiment, we observed the adhesiveness of cells treated with siRNAs to a confluent layer of cells treated with the same siRNAs (Figure 4g). Interestingly, we could observe a significant increase of intercellular adhesiveness upon *endoA3* depletion, as compared to control condition. These variations of intercellular adhesion correlate with the variations of CD166 abundance at the cell surface.

In other words, upon *endoA3* depletion, the higher amount of CD166 at cell surface increases intercellular adhesiveness, which results in slower movements and loss of directionality of individual cells. This may explain why gap closure in wound healing assays is strongly impaired in the absence of *endoA3* compared to control condition in U2OS cells. However, in the absence of CD166, the reduced intercellular interactions only results in a loss of directionality, with very minor effect on migration velocity of individual cells. This may explain why gap closure is only slightly affected in these conditions. However, the very strong effect of *endoA3* depletion on cell migration requires the presence of CD166. We conclude that *endoA3*-dependent CIE downmodulates CD166 from cell surface, which reduces intercellular adhesion. Conversely, when *endoA3* is depleted, CD166 accumulates at cell-cell junctions, resulting in stronger intercellular adhesion and reduced migration capacity of cells. Altogether, these observations indicate that fine-tuning of CD166 at cell surface by *endoA3*-dependent CIE can be used by cancer cells to modulate intercellular adhesiveness, and ensure optimal collective migration with the highest speed and directionality.

These new data were added in the indicated figures, and the text was modified accordingly (page 5, line 260 to page 6, line 319).

1) Authors started from a screening looking for proteins whose level was decreased upon clathrin-dependent endocytosis inhibition, claiming that this might be the consequence of upregulation of a CIE pathway leading to degradation. With this argument in

mind, they selected CD166 for their study as it was downregulated upon inhibition of clathrin-dependent endocytosis both at protein and mRNA level. However, in contrast with their initial hypothesis, CD166 internalization is not accelerated by clathrin-KD, AP2-KD or upon other inhibitors, suggesting that CIE is not upregulated. Thus, the issue remains open (and not discussed in the manuscript): why CD166 protein levels are downregulated upon clathrin-mediated endocytosis inhibition? Is this reflecting the transcriptional effect and/or clathrin/AP2 has an impact on CD166 later trafficking steps (not on CIE)? More investigations are needed to clarify this point.

Answer of authors:

The point raised above by the reviewer is very interesting and was also puzzling for us. For this reason, we checked at first place a possible side-effect of AP2 depletion on the transcription of CD166 gene (Extended Data Figure 1e). We observed a 2-fold (50%) decrease of CD166 gene expression by qPCR. However, this could not fully explain the very strong downregulation of CD166 from cell surface measured by proteomic analyses or by flow cytometry (between 3- to 4-fold decrease). This led us to the hypothesis that CD166 was probably endocytosed by a CIE mechanism that could be upregulated. This hypothesis appeared to be wrong, as correctly observed by the reviewer.

The alternative hypothesis is actually that AP-2 depletion has an impact on post-endocytic trafficking steps of CD166, and consequently on the fate of the protein. The rationale for this alternative hypothesis was already proposed in the literature for other CIE cargoes by Donaldson's lab (Dutta, D., Donaldson, J.G. 2015. Sorting of Clathrin-Independent Cargo Proteins Depends on Rab35 Delivered by Clathrin-Mediated Endocytosis. Traffic. 16(9):994-1009). In this paper, authors clearly show a post-endocytic crosstalk between CIE and CME, and decipher molecular aspects of this crosstalk. Some CIE cargoes, such as CD44, CD98 and CD147, are normally recycled back to plasma membrane via recycling endosomes in an Arf6-dependent manner and avoid degradation. However, upon AP-2 depletion, these cargoes are re-routed to EEA1-positive endosomes to finally reach lysosomes, where they are degraded.

[REDACTED]

[REDACTED]

The post-endocytic fate of CD166 remains a very interesting question and will require further investigations. However, for a sake of clarity, we believe that these data are out of the scope of the present manuscript and should be part of another story focusing on later steps of CD166 trafficking. Nevertheless, we added a sentence in the manuscript to clarify this hypothesis (page 2, line 89-92).

2) The data on dynamin is not convincing. There's a clear effect at 5 and 10 min (more than 50% magnitude of inhibition), exactly when early steps of internalization are measured. I understand that this is not statistically significant, but more statistics is needed to exclude a role.

Answer of authors:

The reviewer is referring here to the kinetics of anti-CD166 endocytosis measured by loss of surface assays in flow cytometry (Figure 1b). We understand the reviewer's criticism. However, in the light of the dataset obtained by another approach presented in Figure 1c

(continuous uptake measured from confocal images), dynamin depletion does not seem to affect significantly the uptake of CD166. Nevertheless, we decided to redo those experiments in better controlled conditions (fresh medium, better control of pH...), as the data presented in the previous version of the manuscript showed a lot of variability. These new data – statistically more reliable – are now replacing the former graph in Figure 1b and still do not show any significant effect of dynamin depletion on CD166 uptake.

3) The physiological impact of the pathway on CD166-dependent migration and immunogenic response is preliminary. Authors should provide stronger molecular genetic evidences for a causal link with Endophilin A3/Galectin-8 mechanism.

Answer of authors:

We totally agree with the reviewer's point of view. For our detailed answer, please see above (point C of our answer to his/her general comments).

4) Minor point: note that Endophilin A1, A2 and A3 are not “isoforms” as they are products of different genes. The nomenclature along the manuscript is wrong and need to be revised.

Answer of authors:

Although we understand the point of view of the reviewer, we used the terminology "isoform" in accordance with the common usage in the field for endophilin-A proteins. In addition, according to the MeSH (Medical Subject Headings) of the National Library of Medicine (NLM) of NIH (Bethesda, US), protein isoforms are defined as "different forms of a protein that may be produced from different GENES, or from the same gene by ALTERNATIVE SPLICING" (<https://www.ncbi.nlm.nih.gov/mesh/68020033>). MeSH is the NLM controlled vocabulary thesaurus used for indexing articles for PubMed. For these reasons, we think the nomenclature used in our manuscript is not wrong and widely accepted in the community. We decided to keep it as in the initial version of the manuscript, unless the reviewer and/or the editor provide referenced arguments that would justify a modification.

Reviewer #3 (Remarks to the Author):

In this manuscript, the authors identify novel cargos of the Non-Clathrin-dependent endocytic (NCE) pathways. They, in particular, focus on CD166 (ALCAM) and provide a set of convincing evidence that this cargo enters an NCE route that utilizes endophilins as membrane curvature, BAR-containing determinant. Importantly, they specifically show that Endophilin-A3 mediates the internalization of CD166 in various cell types. Collectively, these results support the existence of cargo and adaptor-specific entry routes in NCE. This study, in particular, challenges the prevailing view that the three endophilin-A isoforms are functionally redundant in NCE. This is interesting. However, while the data presented do support this contention and further enable the authors to identify a specific cargo, CD166, that uses EndoA3, there is little effort to understand where the specificity comes from and neither suggestions nor experiments as to the possible mechanisms underlying the specificity of the different Endophilins for different cargos, different vesicles and distribution pattern. A number of key questions remain unaddressed. For example, does EndoA3 interact with CD166? Can a structure function analysis of the 3 endophilins reveal insights into how they might act and in an ostensibly specific fashion? These and a number of issues underlined below weaken the enthusiasm for this manuscript and question whether the findings reported are sufficient to justify publication in NC.

Answer of authors:

We would like to thank reviewer #3 for his/her meaningful comments, that greatly helped us to improve the manuscript. His/her main concern was about the mechanisms that regulate the specificity of endoA3 for CD166 cargo. In particular, he/she would like to know whether endoA3 interacts physically with CD166. To address this question, we performed pull-down experiments from HeLa cell lysates using the cytosolic fragment of CD166 (fused to GST, produced in bacteria and bound to glutathione resin). Interestingly, we could pull down endoA3, but not endoA2. Of note, we could also pull down ezrin, used here as a positive control. Interaction of ezrin with the cytosolic tail of CD166 has previously been documented (Tudor, C., et al. 2014. Syntenin-1 and Ezrin Proteins Link Activated Leukocyte Cell Adhesion Molecule to the Actin Cytoskeleton. J Biol Chem. 289(19):13445-13460). This physical interaction of endoA3 with CD166 may explain the specificity of the endocytic mechanism. These data were added in the manuscript in Figure 2d and the main text was modified accordingly (page 3, lines 150-157). We cannot exclude that specific features in the sequence/structure of each endoA isoform play a role in the specificity of their action. This question is very interesting, but requires further investigations that go beyond the scope of this manuscript. We are currently working on this aspect, that will be the topic of a future study.

Specific points

1. The authors start by a mass spectrometry approach to identify cell surface proteins that are altered upon inhibition of Clathrin-mediated endocytosis (CME), and focus specifically on CD166, the levels of which are severely reduced after Clathrin ablation. This data is interesting but the logical premises would need to be clarified. Specifically, it is unclear as to why the authors decided to focus on proteins that display a reduced cell surface levels after inhibition of Clathrin in order to identify non-Clathrin endocytic cargos. In

the most obvious scenario, perturbation of CME should not affect the entry of cargo that use a different entry route, yet they end up studying CD166 whose cell surface levels are severely reduced following Clathrin inhibition.

Answer of authors:

*We thank the reviewer for this important comment. Reading again the §2 of page 2, we noticed a mistake in the following sentence, that we readily corrected in the revised version of the manuscript: "In order to discover new clathrin-independent cargo proteins acting in cancer, we performed a quantitative proteomic screening for proteins whose cell surface exposure was **unchanged or even decreased** upon inhibition of CME (...)". The idea behind this analysis was that CME canonical cargoes should accumulate at cell surface when CME is inhibited, while CIE cargoes are expected to remain stable or even decrease. Indeed, we observed an accumulation of canonical CME cargoes at cell surface (Tf and LDL receptors), while CIE cargoes (e.g. CD44) or cargoes that use both CME and CIE (e.g. EGFR) were not be affected. However, to our great surprise, we observed that some proteins were strongly downregulated from cell surface, CD166 being the most extreme case. We first hypothesized that CME inhibition might upregulate some CIE processes. However, our experiments (Figure 1a,c, Extended Data Figure 2a,b) did not show any acceleration of anti-CD166 uptake upon CME inhibition. For this reason, we checked at first place a possible side-effect of AP2 depletion on the transcription of CD166 gene (Extended Data Figure 1e). We observed a 2-fold (50%) decrease of CD166 gene expression by qPCR. This decrease could not fully explain the very strong downregulation of CD166 from cell surface measured by proteomic analyses or by flow cytometry (between 3- to 4-fold decrease). All these observations led us to an alternative hypothesis where AP-2 depletion could impact the post-endocytic trafficking steps of CD166, and consequently the fate of the protein. The rationale for this alternative hypothesis was already proposed in the literature for other CIE cargoes by Donaldson's lab (Dutta, D., Donaldson, J.G. 2015. Sorting of Clathrin-Independent Cargo Proteins Depends on Rab35 Delivered by Clathrin-Mediated Endocytosis. *Traffic*. 16(9):994-1009). We invite reviewer #3 to read our answer to specific point #1 raised by reviewer #2, that should further clarify this hypothesis. We also added a sentence in the manuscript to clarify this hypothesis (page 2, line 89-92).*

2. As comment to figure 1d, it is stated that: "In addition, continuous co-incubation of cells with fluorescently labeled anti-CD166 antibody and transferrin clearly showed that both cargoes do not co-localize (Figure 1d)." While the data do support a lack of colocalization between these two cargoes, their interpretation as indication of the fact the cargoes under scrutiny enter via distinct routes is unwarranted. Indeed, these experimental results are surprising as one would expect that after entry into cells some levels of colocalization between NCE and CME cargoes in either early or late endosome should be detected. The lack of such colocalization does requires some explanation and should not be used as indication that CD166 employs an entry route different from CME. A technical, but relevant issue to be addresses in this context is whether after continuous incubation with anti-CD166 and anti-TFR the surface bound abs have been completely removed (by acid washes) to ensure that only internalized abs are followed. In the methods section, 3 rapid (20") acid washes are described but it is unclear whether experimentally the authors verified that after this treatment CD166 antibodies had been removed from the cell surface.

Answer of authors:

We thank the reviewer for these important remarks, and here is our answer:

A) Continuous co-uptake of fluorescent Tf and anti-CD166 presented in this figure was performed for 5 minutes. As the kinetics of Tf and anti-CD166 uptake are very different (see Figure 1a: Tf uptake is much faster than anti-CD166), we do not expect that they reach the same final compartment after such a short time. In addition, as incubation with Tf and anti-CD166 were performed continuously, we expected to saturate the endocytic structures from plasma membrane all along the way to the endocytic compartment reached after 5 min for both cargoes. In this case, we would have expected at least a partial overlap of both signals if both cargoes strictly used CME, even if kinetics are very different. That is not what we observed.

B) Nevertheless, we agree with the reviewer that this piece of data should not be taken alone as a formal proof of distinct entry routes for the two cargoes, but rather as an additional indication. It has to be considered together with uptake assays that come before (Figure 1a-c, Extended Data Figure 2a-h) and lattice light-sheet microscopy data that come after (Figure 1e-h, Extended Data Figure 2i) in the manuscript. We do not have the impression that we overinterpret this data in the corresponding paragraph of the main text (page 2, lines 92-94).

C) The acid wash protocol we used was previously published by Boucrot et al. (2015) and Chan Wah Hak et al. (2018) (see references of the main manuscript), where it was used with great success for different antibodies. Of course, we verified experimentally if this protocol was efficient to strip off anti-CD166 antibody from cell surface in our hands. We performed cell surface staining with anti-CD166 and looked if residual signal was remaining after this acid wash protocol. Immunofluorescence images showed very little remaining signal. This was quantified by flow cytometry, and we obtained >95% removal of cell surface bound anti-CD166 antibody. A sentence claiming the efficiency of acid washes was added in the methods section (page 38, lines 1299-1301).

3. In Figure 2A and Suppl-3A, it is shown that ablation of Endophilin A3, but not of A1 or A2 impairs the internalization of CD166. However, in Fig. 2A what is measured is the relative amount of internalized CD166 with respect to mock transfected cells, whereas in Suppl-3A the relative levels of cell surface CD166 are reported. This second measurement indicates that inhibition of Endo-A3 only marginally reduces CD166 surface levels (from ~ 40% to ~ 30%), suggesting Endo-A3 is, at best, partially responsible for CD166 entry and other routes must be at play. Hence, caution in interpreting these results should be used. These findings also beg the question as to whether the concomitant inhibition of Clathrin and EndoA3 impacts more dramatically on CD166 surface levels and endocytosis.

Answer of authors:

A) First, we would like to clarify an important point so that things are not technically misleading. Extended Data Figure 3a does not show the relative levels of CD166 at cell surface, but rather the percentage of endocytosed CD166 after 40 min (measured by loss of surface relatively to the 0 min time point, that corresponds to 0% of endocytosis).

B) Experiments presented in Figure 2a and Extended Data Figure 3a cannot be quantitatively compared, as the experimental conditions are completely different (for more details, please see our very detailed Methods section). Figure 2a was obtained from a continuous incubation of cells with anti-CD166 antibody at 37°C for 10 to 30 min, without any binding step on ice. Extended Data Figure 3A was obtained from an incubation of cells on ice for 30 min with anti-CD166 antibody to label cell surface accessible cargo, which were then switched to 37°C for 40 min. In addition, Figure 2a was obtained from quantification of accumulated signal inside the cells from confocal images, while Extended Data Figure 3a was obtained from loss of surface measured in flow cytometry. Although data cannot be quantitatively compared between the two types of experiments, qualitative observations show similar trends.

C) However, we agree with the reviewer that results are not "black or white". As mentioned in the discussion of the data in the manuscript, there are most probably other BAR domain proteins and/or other galectins that are involved in the endocytosis of CD166, in the same or in parallel endocytic routes. This remains to be explored. In addition, as suggested by the lattice light-sheet microscopy data, CME could also account for the residual endocytosis of CD166.

D) Inhibition of CME alone is not sufficient to observe a significant effect on CD166 uptake (see Figure 1a,c; Extended Data Figure 2a-b). Nevertheless, we performed the experiments suggested by the reviewer, combining depletion of clathrin and endoA3 (Extended Figure 3f). In our experimental conditions (similar to Figure 2a), depletion of clathrin did not affect further the uptake reduction caused by the depletion of endoA3.

4. The Plot in figure 2b that correlates the levels of endocytosis with Endo A1, A2 and A3 expression is redundant and add little information to the data reported in 2A.

Answer of authors:

We do not totally agree with the reviewer on this point. Figure 2a does not contain the information regarding the relative expression level of each endoA isoform in the various depletion conditions. We imagined that the expression of a specific isoform could be affected by the depletion of others. This allowed us to extract a more precise correlation between the endocytic level of CD166 and the expression level of each isoform.

5. "In addition, overexpression of endo-A3 – but not endo-A2 – significantly increases the uptake of CD166 (Extended Data Figure 3c)." It is unclear how to interpret these results. At face value, they would suggest that Endo-A3 is limiting in determining the entry of CD166. However, to corroborate this conjecture it would seem necessary to restore the expression of EndoA3 in those cells where its levels are low or undetectable, such as SUM159, LB33-MEL and MZ2-MEL.43 cells, and test CD166 internalization.

Answer of authors:

We thank reviewer #3 for this great idea. We re-expressed endoA3 in LB33-MEL cell line and observed a nice restoration of CD166 uptake (Extended Data Figure 3e).

6. “Interestingly, TIRF imaging of living cells showed that GFP-labeled endoA3 has a very short life-time at cell surface, as ~90 % of spots lasted for less than 30 seconds (Extended Data Figure 5e-g, green bars on the histogram). Interestingly, endoA3 became much less dynamic at sites of CD166 clustering, as ~80 % of endoA3 spots had a life-time exceeding 50 seconds and often the total duration of our 120 sec movies (Extended Data Figure 5e-g, red bars on the histogram; Supplementary material, Movie 6)”

Here, the authors used TIRF to study the dynamics of CD166 and EGFP-EndoA3. It would be good to use similar TIRF-based analysis to study CD166 dynamics in the absence of Endo-A3 and upon its overexpression. Indeed, the authors points that at sites of EndoA3 clustering the dynamics of CD166 is robustly slowed down. This is apparently in contrast with the findings reported above that elevation of EndoA3 increases CD166 internalization and some explanation is required.

Answer of authors:

The idea proposed here by the reviewer is very interesting. Before going further, we would like to clarify one point: we observed that at sites of CD166 clustering, the dynamics of endoA3 is robustly slowed down (and not the contrary). However, as proposed by the reviewer, we tried to see in TIRF experiments if dynamics of anti-CD166 antibody bound to cell surface is modified in the absence of endoA3. However, the data obtained were not conclusive. The main reason is that among all cell surface structures labeled with anti-CD166 antibody, we cannot strictly distinguish those that are (pre-)endocytic from those that are non-endocytic. Hence, we could not extract any differences on anti-CD166 dynamics with and without endoA3.

The change in endoA3 dynamics when it is associated with the cargo is not totally surprising, as a similar behavior was already observed for endoA2 upon association with cargoes such as ligand-stimulated beta-adrenergic receptors or STxB (see main text of the manuscript, page 3, lines 148-150).

7. The authors provide evidence that EndoA1, A2 and A3 display distinct cellular localization patterns. The data are overall convincing but their outcome could be strongly influenced by the sensitivity of the antibody used or the levels of expression of EndoA3-GFP. Some cautionary wording would be welcomed. Alternatively, an EM study of the various Endophilin isoforms might address whether they are indeed distinctly localized.

Answer of authors:

We thank the reviewer for this important comment. We forgot to mention that we tested the specificity and sensitivity of our antibodies before using them. Basically, we depleted their target protein with siRNAs and checked their efficiency in both Western blot and immunofluorescence. To make this important point clear to the reader, we added a statement in the Methods section, Antibodies and other reagents (page 34, lines 1060-1061). In addition, in the absence of any functional anti-endoA3 antibody, we expressed GFP-tagged endoA3. We took care of avoiding too strong overexpression in our experiments, as already mentioned in the text of the main manuscript (page 4, lines 176-177).

8. 10 mM LacNAc treatment is shown to inhibit robustly CD166 entry, to a significantly larger extent with respect to Endo-A3 ablation. This might be due to a number of differences in the two experimental conditions. Nevertheless, one would expect that after Endo-A3 inhibition, when a sizable fraction of CD166 still enters the cells as judged by the data reported in S3A, the addition of 10 mM LacNAc should not further reduce CD166 entry if, as proposed, the Endo-A3 endocytic route is galectin-dependent. Similar experiments should also be performed using siRNA against Galectin 8

Answer of authors:

As correctly pointed out by the reviewer, the comparison of endocytic level between these two experiments is risky. The main difference between the two is the absence of serum in LacNAc treatment experiment (Extended Data Figure 9a). In the uptake experiment where endoA3 is depleted, serum is present and many factors that it contains may affect the absolute level of endocytosis (Figure 2a). Among these factors, there are of course numerous galectins. To summarize, we would say that:

A) Owing to the experimental differences, we cannot say that LacNAc treatment inhibits CD166 uptake to a significantly larger extent than endoA3 depletion.

B) As already proposed in our answer to point #3, alternative endocytic routes may explain the residual uptake of CD166 in our data. Things are definitely not black or white. In particular, with LacNAc treatment, we inhibit the binding of galectins, which may strengthen inhibition of uptake if, as proposed in our discussion, other galectins than Gal8 are involved in the process.

C) We performed new experiments in the absence of serum, that are more comparable to LacNAc experiment. Very interestingly, we observe that the effect of endoA3 depletion on anti-CD166 uptake is stronger in these conditions (~70% decrease, Extended Data Figure 9h) compared to the results presented in Figure 2a that were obtained in the presence of serum (~33% decrease). Hence, depletion of endoA3 (~70% decrease) or LacNAc treatment (~62% decrease) do not show significant differences. What could explain those differences between the presence and the absence of serum? As proposed earlier, serum brings many nutrients that could stimulate alternative endocytic mechanisms (many galectins, growth factors such as Tf that could stimulate CME, etc). As proposed by the reviewer, we also combined the depletion of endoA3 and Gal8 on the uptake of anti-CD166 antibody (see Extended Data Figure 9h). However, we did not observe any significant additive effect, as depletion of endoA3 already reduces anti-CD166 uptake to a low level. This also suggests that endoA3 and Gal8 are functionally involved in the same mechanism.

9. In figure 4, the authors attempt to broaden the meaning of EndoA3-dependent NCE exploring the impact on tumoral cell migration and T cell proliferation. They initiate by showing that EndoA3 siRNA impairs wound closure of U2OS, only when these cells express CD166. This is an elegant experiment that, however, would need to be strengthened by the use of additional cancer cells, and specifically of tumor cells that do not express EndoA3, but still possess CD166. It would also be necessary to test CD166 internalization in U2OS and to test the impact of EndoA3 ablation on wound healing in HeLa cells.

In general, the attempts to link Endo-A3 to cancer biology appears weak, at best preliminary precluding the authors to draw any meaningful conclusions as to the role of Endo-A3 in cancer biology, and specifically on cancer cell migration/ invasion.

Answer of authors:

We agree with the reviewer that the data exploring the impact of endoA3-dependent CIE are still preliminary. Unfortunately, in our experimental conditions, we could not observe any significant change in the proliferation of T cells when endoA3 was depleted from cancer cells. We think that going deeper into the role of endoA3-mediated CIE of CD166 on immunogenicity of cancer cells will require to investigate new readouts, and that goes beyond the scope of the present study. In order to preserve the clarity and maintain the homogeneity of the manuscript, we have decided to remove this preliminary data, in agreement with the editor. However, we added new data that strengthen the role of endoA3-mediated CIE of CD166 on cancer cell migration, and we hope the reviewer will appreciate them (see below).

A) As requested by the reviewer, we performed wound healing assays on other cancer cells lines: wild-type SUM159 breast cancer cell line and a genome-edited clone of this cell line expressing AP2-GFP (named SUM159 AP2-GFP). While both cell lines are negative for endoA3, they show different patterns regarding CD166 expression: CD166 is undetectable in Western blot in SUM159, while it is strongly expressed in SUM159 AP2-GFP clone (see blots in Extended Data Figure 10g). We sought to use those differences to our advantage in wound healing assays. Interestingly, the treatment with siRNAs against endoA3 did not affect gap closure, as the protein is not expressed. In addition, depletion of CD166 was efficient (see blots in Extended Data Figure 10g), but did not affect significantly gap closure. This data clearly indicates that the strong effect on gap closure observed in U2OS upon endoA3 depletion is directly connected to the expression/function of the protein, and is not an off-target effect of siRNA treatment. Of note, we did not investigate gap closure by wound healing on HeLa cells, as they have very poor migration abilities.

B) As asked by the reviewer, we checked the effect of each endoA isoform depletion on anti-CD166 uptake in U2OS cells (Extended Data Figure 3c). As for HeLa cells, CD166 uptake was only reduced upon endoA3 depletion. Very interestingly, we also observed an increase of cell surface CD166 level in U2OS upon endoA3 depletion by flow cytometry measurements, as for HeLa cells (Extended Data Figure 4g). Altogether, these data show that U2OS behave similarly to HeLa, and that the regulation of CD166 abundance at cell surface by endoA3-dependent CIE is a general mechanism shared by cancer cells that express both proteins.

C) We added new analyses and data strengthen the role of endoA3-mediated CIE of CD166 on cancer cell adhesion and migration, and we hope the reviewer will appreciate them. For more details, we convey reviewer #3 to read our answer to the general comments of reviewer #2 (more precisely point C).

REVIEWERS' COMMENTS:

Reviewer #1 (Remarks to the Author):

The authors have addressed all of my concerns. The manuscript has been significantly improved and is ready to be published in Nature Communications. Nevertheless, I do have one more suggestion: given the diversified functions of endophilin A family members reported in published researches and the current study, I agree with review #2 and would suggest the authors replace "isoforms" with "family members" in reference to this group of proteins.

Reviewer #2 (Remarks to the Author):

The revised version of the manuscript has significantly improved. Authors have performed additional experiments addressing almost all my major concerns.

I agree with the authors that the immunological data were too preliminary, while the migration data has now improved and provide important insights into the role of CD166 endocytosis via Galectin8/EndoA3.

Although the manuscript still lacks some mechanistic aspects, it is of interest for the cell biology and membrane trafficking community as it describes a novel CIE pathway and its potential relevance for the regulation of cell migration and cell-to-cell adhesion.

Reviewer #3 (Remarks to the Author):

The authors clarified a number of issues raised, some of the logic of the manuscript, and performed additional experiments.

The identification of the role of ENDOA3 on CD166 is novel and overall the data are supportive of the contention stated.

Two main points should be further discussed.

1. The mechanisms through which EndoA3 acts specifically on CD166 internalization. To this end, the authors showed a pull-down experiment using the recombinantly-purified, cytosolic tail of CD166. They found interaction with EndoA3, but not with EndoA2. This is interesting, but tell us a little as to the mechanisms of action of EndoA3 on CD166, whether the interaction is functional (which would require mapping of the surfaces and generation of mutant disrupting the association). Additionally, the extent of this interaction is abysmally low (judging from the input-less that ~0.15% of the total proteins loaded in a cytosolic extract interact with CD166 tail). This might reflect the transient or indirect nature of this association (albeit the author stated that there is a "tight association of endoA3 with CD166-containing endocytic structures" line 152), or that the association is not really specific. Admittedly, at this stage of reviewing defining the nature of the interaction surfaces might pertain to a new installment. Yet the authors should explicitly state that the extent of association in this pull-down experiment is very weak and that the molecular mechanism through which ENDOA3 control CD166 internalization remains the subject of further investigation.

2. The data on the impact of EndoA3 and Cd166 on cell migration has been improved. Particularly elegant are the experiments showing that ENDOA3 silencing impact migration only in CD166 expressing cells. However, all the experiments are strictly done using relatively simple in vitro assays of cell migration and the consequences on tumor growth and progression are totally untested. Thus, while the assays were conducted in tumor cells, it seems that ENDOA3/CD166 axis might impact on sheet migration regardless of the oncogenic state of the cells. The emphasis on tumor biology should probably be toned down.

-The model for the impact of ENDOA3 and CD166 on cell migration also calls for some clarifications. The authors propose that following ENDOA3 inhibition CD166 remains at the cell surface and specifically at cell-cell junction causing cells in wounded monolayers to have reduced

speed and directionality (why directionality, as stated by the authors in the author in their reply to rev#2, should be affected when cell-cell interaction is augmented is not at all obvious). In this type of assays, however, where collective motion is being monitored the tightness of cell-cell interaction has been shown to promote rather than inhibit sheet migration (one obvious mechanism is that cell coordination and the transmission of supracellular forces are favored by the tight cell-cell association). Also, the author uses SUM159 expressing or not AP2-GFP that displays very different levels of CD166. Yet the migration of these two distinct cells is not very different while one would have expected that the AP2-GFP cells, which possess very high levels of CD166, should display decreased migration possibly as consequences of tight cell-cell junctions. Is CD166 not accumulated at the cell-cell surface in these cells? It is also unclear if one of the mechanisms whereby ENDOA3 influences cell migration is by controlling the levels of junctional CD166, why removal of CD166 or its overexpression has a marginal impact on wounded migration. Possibly better assays to lend credence to the model are scattering assays or a wetting/spreading assays.

Point-by-point response to reviewers' comments

Reviewer #1 (Remarks to the Author):

The authors have addressed all of my concerns. The manuscript has been significantly improved and is ready to be published in Nature Communications. Nevertheless, I do have one more suggestion: given the diversified functions of endophilin A family members reported in published researches and the current study, I agree with review #2 and would suggest the authors replace "isoforms" with "family members" in reference to this group of proteins.

Answer of authors:

We thank the reviewer #1 for his/her positive feedback. Regarding this last comment, as we answered to reviewer #2, we used the terminology "isoform" in accordance with the common usage in the field. In addition, according to the MeSH (Medical Subject Headings) of the National Library of Medicine (NLM) of NIH (Bethesda, US), protein isoforms are defined as "different forms of a protein that may be produced from different GENES, or from the same gene by ALTERNATIVE SPLICING" (<https://www.ncbi.nlm.nih.gov/mesh/68020033>). MeSH is the NLM controlled vocabulary thesaurus used for indexing articles for PubMed. Hence, we think that calling endophilin-A family members "isoforms" is appropriate. We don't see why the diversity of endophilin-A functions would justify such a change. There exist many enzymes whose various isoforms have diverse functions, and yet they are called isoforms.

Reviewer #2 (Remarks to the Author):

The revised version of the manuscript has significantly improved. Authors have performed additional experiments addressing almost all my major concerns.

I agree with the authors that the immunological data were too preliminary, while the migration data has now improved and provide important insights into the role of CD166 endocytosis via Galectin8/EndoA3.

Although the manuscript still lacks some mechanistic aspects, it is of interest for the cell biology and membrane trafficking community as it describes a novel CIE pathway and its potential relevance for the regulation of cell migration and cell-to-cell adhesion.

Answer of authors:

We thank the reviewer #2 for his/her positive feedback!

Reviewer #3 (Remarks to the Author):

The authors clarified a number of issues raised, some of the logic of the manuscript, and performed additional experiments.

The identification of the role of ENDOA3 on CD166 is novel and overall the data are supportive of the contention stated.

Answer of authors:

We thank the reviewer #3 for his/her positive appreciation of our revised manuscript.

Two main points should be further discussed.

1. The mechanisms through which EndoA3 acts specifically on CD166 internalization. To this end, the authors showed a pull-down experiment using the recombinantly-purified, cytosolic tail of CD166. They found interaction with EndoA3, but not with EndoA2. This is interesting, but tell us a little as to the mechanisms of action of EndoA3 on CD166, whether the

interaction is functional (which would require mapping of the surfaces and generation of mutant disrupting the association). Additionally, the extent of this interaction is abysmally low (judging from the input-less that ~0.15% of the total proteins loaded in a cytosolic extract interact with CD166 tail). This might reflect the transient or indirect nature of this association (albeit the author stated that there is a "tight association of endoA3 with CD166-containing endocytic structures" line 152), or that the association is not really specific. Admittedly, at this stage of reviewing defining the nature of the interaction surfaces might pertain to a new installment. Yet the authors should explicitly state that the extent of association in this pull-down experiment is very weak and that the molecular mechanism through which ENDOA3 control CD166 internalization remains the subject of further investigation.

Answer of authors:

We totally agree with this comment of reviewer #3 and toned down our claims. As requested, we improved the text of our manuscript by clearly specifying that:

1°) the interaction between CD166 C-terminal tail and endoA3 is weak and likely reflects a transient association (please see the modifications in the main manuscript, lines 178-179, page 4);

2°) the mechanism through which endoA3 controls CD166 uptake requires further investigation. In particular, mapping of interaction surfaces and generation of mutants disrupting this association are necessary to get further insight into this mechanism (please see the modifications in the main manuscript, lines 360 to 363, page 7).

In addition, we toned down one claim, now at line 175 (page 4): "This tight association (...)" was replaced by "This **specific** association (...)". Indeed, the data presented just before this sentence address the question of the specificity of this interaction, rather than its "strength". Hence, we agree that the adjective "tight" was not appropriate.

2. The data on the impact of EndoA3 and Cd166 on cell migration has been improved. Particularly elegant are the experiments showing that ENDOA3 silencing impact migration only in CD166 expressing cells. However, all the experiments are strictly done using relatively simple in vitro assays of cell migration and the consequences on tumor growth and progression are totally untested. Thus, while the assays were conducted in tumor cells, it seems that ENDOA3/CD166 axis might impact on sheet migration regardless of the oncogenic state of the cells. The emphasis on tumor biology should probably be toned down.

Answer of authors:

We agree with the reviewer and toned down our claims:

- We erased the following sentence from our manuscript: "These findings lead to the hypothesis that cancer progression could be directly impacted by the expression of specific clathrin-independent endocytic machineries in cells.";
- We toned down tumor biology aspects in the last paragraph of our discussion (lines 381 to 385, page 8) and put emphasis on the fact that these aspects of the work remain to be done in the future.

-The model for the impact of ENDOA3 and CD166 on cell migration also calls for some clarifications. The authors propose that following ENDOA3 inhibition CD166 remains at the cell surface and specifically at cell-cell junction causing cells in wounded monolayers to have reduced speed and directionality (why directionality, as stated by the authors in the author in their reply to rev#2, should be affected when cell-cell interaction is augmented is not at all obvious).

- We have not elucidated yet why directionality is affected when cell-cell interaction is increased. This aspect will require further investigations.

In this type of assays, however, where collective motion is being monitored the tightness of cell-cell interaction has been shown to promote rather than inhibit sheet migration (one obvious mechanism is that cell coordination and the transmission of supracellular forces are favored by the tight cell-cell association).

- We agree that there is often a correlation between the tightness of cell-cell interaction and migration speed, but this might be valid only in a certain "interaction force window". Our hypothesis is that if cell-cell interaction forces get too strong, they might actually have an inhibitory effect on collective migration. To make things clearer, one can make the analogy with water and its different states. Liquid state is obviously the most favourable situation for collective movements of molecules. In the liquid state, interaction forces between molecules are higher than in a gas state, where these interactions are so weak that molecules move in a disorganized manner. At the opposite, interaction forces are weaker than in the solid state (ice), where these are so strong that flow of molecules is impossible. If we transpose this image to the context of cell migration, we may consider that there must be a "liquid state" of intercellular interactions where forces are optimal to ensure the highest collective migration speed. If cell-cell association is too tight, they might get closer to a "solid state" – where the cell layer would be "frozen" – resulting in slower collective cell movements. The situation where endoA3 is depleted (with increased CD166 abundance at cell-cell junctions) might correspond to such a "frozen state".
- In addition, there is another parameter that should be taken into account and that we did not address yet in our study: the dynamics of cell-cell junctions. We have shown that depletion of endoA3 reduces the uptake of CD166 and increases its abundance at the cell surface. It might also affect the dynamics of CD166 at the cell surface, and consequently the overall dynamics of intercellular junctions. EndoA3-dependent endocytosis might facilitate the redistribution of CD166 during the migration process in order to create new transient junctions. A higher cell-cell junction dynamics might contribute to favor collective cell migration. This hypothesis is currently under investigation and will be addressed in a future study.
- Finally, we should also bear in mind that other endoA3-dependent cargoes might be involved in intercellular interactions. The identification of such cargoes in future studies will likely bring additional light on these migratory aspects.

Also, the author uses SUM159 expressing or not AP2-GFP that displays very different levels of CD166. Yet the migration of these two distinct cells is not very different while one would have expected that the AP2-GFP cells, which possess very high levels of CD166, should display decreased migration possibly as consequences of tight cell-cell junctions. Is CD166 not accumulated at the cell-cell surface in these cells?

- We think that the migration behavior of wild-type SUM159 and SUM159 AP2-GFP cannot be compared, as they are distinct cell lines. Although the AP2-GFP cell line originates from the wild-type SUM159, it is a distinct clone with its own properties. The two cell lines have different levels of CD166, but they are also likely to have very different levels or activity for many other proteins important for migration, such as cadherins, integrins, cytoskeleton, etc. This may explain why we noticed significant differences between the two cell lines in cultures (different spreading, adhesion, cell shape, growth speed...). Hence, we are very cautious and avoid doing such direct comparisons.

It is also unclear if one of the mechanisms whereby ENDOA3 influences cell migration is by controlling the levels of junctional CD166, why removal of CD166 or its overexpression has a

marginal impact on wounded migration. Possibly better assays to lend credence to the model are scattering assays or a wetting/spreading assays.

- Removal of CD166 has a marginal impact on migration, because other surface proteins may play a role in intercellular interactions.
- We did not overexpress CD166 *per se*, but we observed that its accumulation at cell-cell junctions when endoA3 is depleted increases cell-cell adhesion and strongly reduces collective migration. Here, the effect is not marginal at all.
- As explained above, a parameter that we did not explore yet is the dynamics of CD166 at cell-cell junctions, and more generally the impact of endoA3-dependent endocytosis on cell-cell junction dynamics. Hence, the abundance of CD166 is important, but the dynamics of its uptake from (and recycling to) cell surface might also explain these observed effects. This hypothesis is currently under investigation and will be addressed in a future study.